# Growing Inequality in the Coffee Global Value Chain: A Complex Network Assessment

Rebeca Utrilla-Catalan [1], Rocío Rodríguez-Rivero [2], Viviana Narvaez [1,3], Virginia Díaz-Barcos [4], Maria Blanco [5] and Javier Galeano [6,*]

1 ETSIAAB, Universidad Politécnica de Madrid, 28040 Madrid, Spain; rebeca.ucatalan@alumnos.upm.es (R.U.-C.); viviana.narvaez@tecnicafe.co (V.N.)
2 Department of Organization, Business Administration and Statistics, ETSI Industriales, Universidad Politécnica de Madrid, C/José Gutiérrez Abascal 2, 28006 Madrid, Spain; rocio.rodriguez@upm.es
3 TECNICAFE, Cajibío 190518, Cauca, Colombia
4 Department of Química y Tecnología de Alimentos, ETSIAAB, Universidad Politécnica de Madrid, Avda. Puerta de Hierro 4, 28040 Madrid, Spain; virginia.diaz@upm.es
5 CEIGRAM, Universidad Politécnica de Madrid, 28040 Madrid, Spain; maria.blanco@upm.es
6 Complex System Group, Universidad Politécnica de Madrid, 28040 Madrid, Spain
* Correspondence: javier.galeano@upm.es

**Abstract:** Following the liberalization of the coffee sector, governance and power balance in the international coffee trade has changed, with associated impacts on livelihoods in producing countries, most of which are middle- and low-income countries. Drawing on trade data for the period 1995–2018, we examine the dynamics and evolution of the international green coffee market to shed light on the re-distribution of value in the coffee supply chain. Data analysis shows that, over the studied period, the green coffee trade has increased considerably while the number of countries with a relevant role in trade has decreased, so that large exporting countries cover a higher share of trade, to the detriment of small exporting countries. We analyzed various properties of the global coffee trade network to provide insight on the relative contribution of countries not only in terms of their export value but also in terms of other selected features. The green coffee trade has gone from being distributed in many traditionally coffee-producing countries to concentrating mainly on the major coffee producers, as well as in some non-producing countries. These changes in the structure of the international green coffee market have led to greater inequality between producing and importing countries.

**Keywords:** global trade network; green coffee; network inequality; complex networks analysis



## 1. Introduction

Coffee is one of the most widespread drinks, with approximately 400 million cups consumed globally each year [1]. According to the International Coffee Organization (ICO), it is also one of the most traded agricultural commodities in the world and a source of income for millions of smallholder farmers, mostly in middle- and low-income countries [2]. In the period 2018–2019, the top ten coffee-producing countries were Brazil, Vietnam, Colombia, Indonesia, Ethiopia, Honduras, India, Uganda, Mexico, and Perú [2].

The coffee value chain is very complex, with a large number of production stages and the high number of actors that intervene along it, from farmers to final consumers [3]. Four types of actors can be identified in the upstream coffee value chain, from input suppliers to the roaster. First, local suppliers provide inputs to smallholder farmers in tropical regions of Latin America, Asia, and Africa. Smallholder farmers cultivate two types of coffee beans (Robusta and Arabica), mostly via labor-intensive practices, and are primarily responsible for the production of 70% of coffee beans globally [2] and the quality of the finished product. Local farmers sell coffee beans to first- and second-level traders, who

in turn bargain with coffee traders. Large green coffee traders and multinational firms operate in international markets, where most coffee is exported as "green" (not roasted). The downstream coffee value chain (from the roaster to consumption) is usually developed in consuming countries [3].

The coffee market is expanding. Global demand for coffee has increased by more than 60% since the 1990s [2], driving the expansion of production and exports. With more than 70% of the production of green coffee being exported, coffee remains an export commodity [2]. The international coffee market has become more complex, with more countries participating in trade, some non-producing countries increasing exports, and trade of processed coffee gaining ground. This has led to an increasing interest in studying their production and how to improve methodologies through processes that are as environmentally friendly as possible.

To what extent has the development of the coffee market benefited the producing countries? The objective of this study is to analyze how the balance of power in the coffee trade network has evolved in recent decades, after the end of the quota system in 1989. For this, we have investigated the international trade network of green coffee for the period 1995–2018 to examine its evolution over time and how the relative importance of countries participating in this market has varied, both for producing and non-producing countries. The use of modern tools for complex network analysis, data mining, and network visualization has enabled to process of coffee trade data and study the structure and dynamics of the international green coffee trade for the studied period.

The rest of this paper is organized as follows: Section 2 revises previous studies on the coffee sector and value chain, showing that the coffee paradox remains alive. Section 3 describes the methodology where the use of complex network analysis is explained. Section 4 presents the main results and their discussion focusing on the major producing/importing countries. Finally, in the last section, the relevant points are outlined, and conclusions are drawn.

## 2. Literature Review

### 2.1. Coffee Trade Regimes

Coffee was one of the first commodities for which control of world trade was attempted. The international coffee trade has been taking place since the beginning of the 20th century, starting in 1902 with the "valorization" process carried out by the Brazilian state of Sao Paulo [4].

Until the Second World War, Brazil centralized the world coffee market Brazil. In the post-war period, control schemes became involved in other Latin American countries as well. Finally, in 1962 the first international coffee agreement (ICA) was signed by most producing and consuming countries [4]. Between 1962 and 1989, the target price for coffee was set by the ICA regulatory system, and export quotas were allocated to each producer, acting on the second as the forecast of the first was modified.

When the ICA system was in force, the coffee chain was not clearly driven by any producing or consuming countries. Entry barriers in farming and domestic trade were often mediated by governments. This system had problems, but most analysts agree that it was successful in raising and stabilizing coffee prices [5–9].

The ICA ended in 1989 and its collapse had different consequences. First, the governance of the system shifted from producing and consuming countries, which shared control of the international coffee trade, to an increasingly "buyer, trader or roaster driven" value chain [4,10]. Second, as part of structural adjustment policies, the national coffee boards it dismantled and the coffee marketing systems were liberalized [11]. According to Ponte [4], the average real indicator price for the period 1990–1993 was only 42% of the average of the final four years of ICA activity (1985–1988). In later years, there was an increase in prices, due to the frosts and drought in Brazil (1994–1995), and the speculative rise of 1997; even so, the composite average price for the period 1994–1997 was still 20% lower than in 1985–1988 [12].

Between the 1970s and 1980s, Talbot [13] showed a substantial transfer of resources from producing to consuming countries, irrespectively of price levels. He estimates that, in the 1970s, an average of 20% of total income was retained by producers, while the average proportion retained in consuming countries was almost 53%. In the 1980s, there were no great changes, producers still controlled almost 20% of total income and consuming countries the 55%. However, after the collapse of ICA in 1989, the situation changed radically. Between the periods 1989–1990 and 1994–1995, the share of total income gained by producers dropped to 13% while it increased to 78% in consuming countries.

In the 1980s and 1990s, investment funds have become increasingly active in commodity markets [4]. The trend-following operating model tends to cause larger movements in and out of the market than if the market was operated by the coffee industry alone [14]. This activity increases price volatility that mainly affects small-scale farmers and traders in producing countries who do not have access to hedging instruments traders in producing countries [7]. The process of coffee commodification has been described as a problem for exporting countries and producers [15].

In the 1990s and 2000s, the introduction of Structural Adjustment Programs (SAPs) saw a transformation in the policy landscape surrounding coffee production. The global market regime was liberalized but price volatility increased in the aftermath of liberalization policies and SAPs.

The result was the weakening of governance structures at the production end, the intensification of vertical integration, and the consolidation of power and profits in the retail segment [16] with a rapid reduction in the producers' shares of the final retail price [4,17,18]. While farmers lost decision-making capacity over farm activities, intermediary actors (wholesalers, roasters, traders) and major multinational food and beverage companies increased concentration and vertical consolidation [19,20]. In the early 2000s (2000–2004), the price of coffee slumped, creating major social problems across coffee-producing regions of the world [18,21].

The world price of coffee is established in accordance with the supply and demand conditions of the item in the world market. The two main markets for coffee beans are located in the New York (Arabic coffees) and London Stock Exchanges (Robusta coffees). Both operate under two modalities: current or physical markets and future contracts. The latter does not involve physical transactions, since purchase-sale contracts are made specifying the aspects related to the quantities, the qualities required, and the delivery times. In 1980, the amount of coffee traded in the futures market was only approximately 4-fold the coffee traded in the physical market. By the early 1990s, the ratio had risen to 11 fold [22].

World coffee production has always been characterized by high instability, with crop fluctuations, high input, knowledge intensive, slow time to maturity of the coffee cherry, necessitating high investments with delayed returns contributing to price volatility [3]. Climate change or new diseases also introduces inherent financial risk for growers [4]. The combination of factors that affect price volatility, crop features and market liberalization, has placed added burden on smallholders who were previously able to make a decent living from coffee production [4,18,19], jeopardizing the sustainability of coffee production systems. The production and trade of these goods constitute the mainstay of the economies of most developing countries, mainly in terms of employment and export earnings, making the economies of the countries that depend on this item very vulnerable and affecting the living conditions of the smallholder producers that participate in the coffee value chain.

There is evidence that drops in coffee prices act as a push factor for internal migration [23]. Declining global commodity prices and low farm gate prices, decrease farm productivity and disincentivize investments in production [24]. Diminishing income from coffee cultivation has been implicated in the abandonment of cultivation by young people, who seek more lucrative employment off farm [25].

### 2.2. Prices Grow but Development Remains Slow: The Coffee Paradox

Many reasons may explain why some developing countries remain in the developing world while other countries are increasing their socio-economic growth. Some authors even blame international development cooperation for perpetuating this situation [26,27]. Sometimes developing countries themselves have governments highly dependent on the export of commodities. These governments focus exclusively on the expansion of production areas instead of promoting agricultural development leading to rural welfare or investing resources in the development of the country [28,29].

It would be expected that when agricultural commodities in developing countries experience an economic boom, they would offer possible ways to escape poverty. Just as expected is the creation of environmental and social problems, which have been studied extensively in recent years, such as environmental problems arising from deforestation [30]. The economic principles of early trade theory assume that a country rich in natural resources would use them intensively. Thus, many of the countries with large areas of forest and arable land export timber and agricultural products, most of them being developing countries, and some studies even find that improved commodity exports increase deforestation in southern countries [31,32].

Among the social problems derived, most studies focus on the global value chain (GVC) and its verticality, based on a "win–lose" relationship in which business always wins and small producers always lose [33]. In another approach, McCarthy et al. [28] pointed out the problems derived from land tenure and the lack of regulation as a reason for social conflicts among producers.

Coffee is one of those commodities whose market is experiencing considerable growth. Coffee is mainly produced in developing countries while it is mainly consumed in industrialized countries, thus representing a vertical or North–South relationship in the value chain. According to statistics from the ICO, in the period 2018–2019, the world consumed more than 168.5 million bags of coffee, and North America, Europe, and Japan accounted for approximately 56% of coffee consumption [2].

In particular, the coffee value chain changed drastically after the deregulation of the coffee trade at the end of the 20th century, which Ponte [4] analyzes in detail. His study demonstrates how by analyzing the global coffee chain it is possible to understand the existence/absence of economic development. Many developing countries depend on coffee exports, being the livelihood of millions of small producers in these countries. Some authors point to the innovation and research as the way to increase the benefits of local producers [34]. On the other hand, local governments are aware of this dependence on exports for local producers and therefore have tried to control and regulate operations. But this control passed into the hands of the large multinationals after the liberalization of coffee. Indeed, in recent decades, as Ponte [4] pointed out, the market has become more concentrated and only the largest coffee traders have survived.

At this point, the well-known "coffee paradox" emerges [10], characterized by decreasing and unstable prices to farmers on the one side and increasing consumer prices on the other side.

The coffee market has turned from a producer-driven market to a buyer-driven commodity supply chain [4,35,36]. Firms operating downstream in the supply chain add value to meet consumer demands, combining different coffees in blends, offering different roast grades, symbolic attributes and services provided in bars and coffee shops, but without involving upstream firms [3,37].

There is indeed growing awareness on the part of consumers, who are increasingly looking for healthier and more socially responsible food [38]. This is leading to establishing more integrated and less vertical value chains [39], to the birth of what is considered inclusive businesses [16], and, above all, to the increase in the certification of products under socially and environmentally sustainable standards and labels. The increase in this type of certifications has also given rise to a growing literature on very diverse products [40,41], and, with much relevance, on coffee [42–44]. However, so far, it has been very difficult to

establish if differences in development between certified and non-certified smallholders are due to certification. For example, Jena et al. [45], in assessing the impacts of coffee certification in Ethiopia, showed that certification contributed to increased incomes among coffee farmers, but the impact on their poverty status was negligible. The same was true for studies by Arnould et al. [46] in Peru, Guatemala, and Nicaragua, and by Chiputwa et al. [43] in Uganda. Recently, Naegele [47] found that the farmer receives approximately one-sixth of the price premium coffee Fair Trade paid by the consumer.

It becomes clear that something more is needed to generate smallholder development. Recent studies by Kopp and Salecker [48] point to access to education and market information, as well as access to microcredit and improved infrastructure, as key to generating development once and for all.

That is why the analysis of the international market through complex networks, which show a global view, could be very useful to incorporate new ideas about economic complexity.

## 3. Materials and Methods

Our study focuses on the international market of green coffee beans, under the Harmonized System HS6 [49], to analyze how it has evolved over the last three decades. The assessment of the global trade network for green coffee has been carried out using bipartite networks, for which we have differentiated two subgroups, one of exporting countries and the other of importing countries, creating a weighted link by the market volume in US Dollars.

In our opinion, the use of complex network analysis for global visualization is a useful tool, since it allows a single image to see what the market for this commodity is like. Many authors support this idea and apply complex networks analysis to study International Trade networks of different commodities [50].

### 3.1. Data

The data used for this study are exclusively data on the dollar amount of green coffee exports from the world's leading exporting countries for the period 1995–2018. In particular, we are studying the volume of money that is exchanged in the international coffee market, not the number of coffee beans that is transported in the world. Due to the variability and enormous amount of data on the green coffee market over the years, data have been selected from countries with a percentage of green coffee exports equal to or greater than 1% of total exports for the period 1995–2018. This value has been chosen as it is considered a sufficiently significant percentage of green coffee exports, considering lower percentages as values with little relevance for market analysis, and also covering with these data approximately 90% of the market corresponding to each year.

The information age requires a new standard of data availability, a concept often referred to as public transparency. Focusing on the availability and free access to all available trade data, the United Nations and its Statistics department facilitate these data by publishing them in the UN Comtrade database (United Nations International Trade Statistics Database) [49]. The UN Comtrade is a highly specialized database; with over 40 billion data records since 1962 is the largest depository of international trade data available publicly on the internet. Some related studies about international trade networks also used this database, see for instance [50].

Over 170 reporter countries provide information about the International Trade Market, all commodity values are converted from national currency into US Dollars using exchange rates, supplied by the reporter countries or derived from monthly market rates and volume of trade. Quantities, when provided with the reporter country data and when possible, are converted into metric units. Commodities are reported in the current classification and revision (HS 2012 in most cases as of 2016). The data are permanently stored in the UN Comtrade database server. In our study, we have focused on level HS6, code 090111, "Coffee; not roasted or decaffeinated".

### 3.2. Preliminary Visualizations

A preliminary analysis has been developed using tools from The Observatory of Economic Complexity, which is a data visualization site for international trade data created by the Macro Connections group at the MIT Media Lab [51].

To provide a more general vision of the evolution of the green coffee trade, graph of accumulated temporal evolution (Figure 1) and treemaps of each year (Figures 2 and 3) are generated.

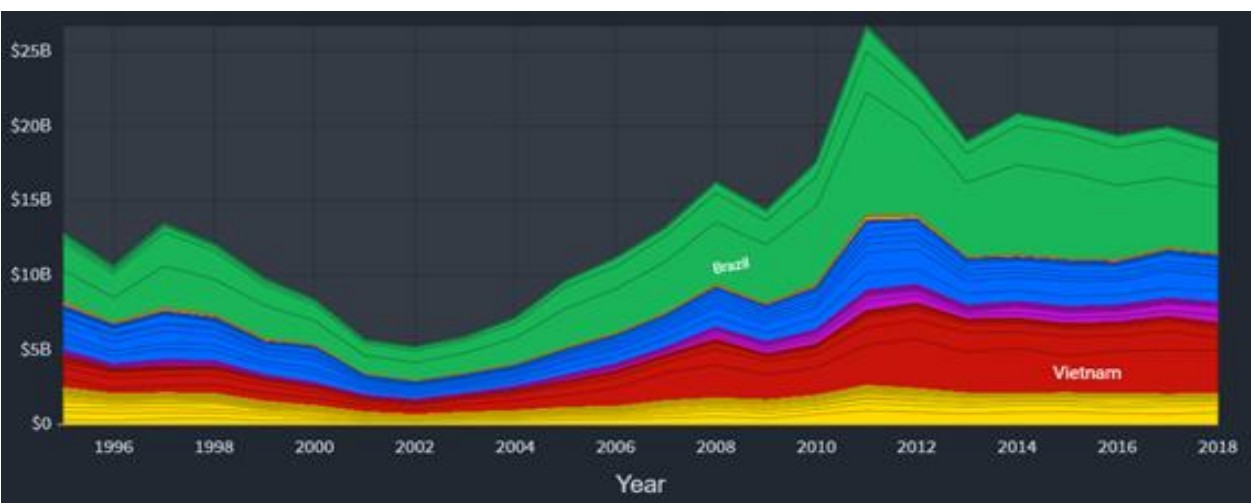

**Figure 1.** Graph of stacked green coffee exports worldwide for the period 1995–2018 in US$. The colors represent different continents: Green (South America), Blue (North and Central America), Red (Asia) and Yellow (Africa).

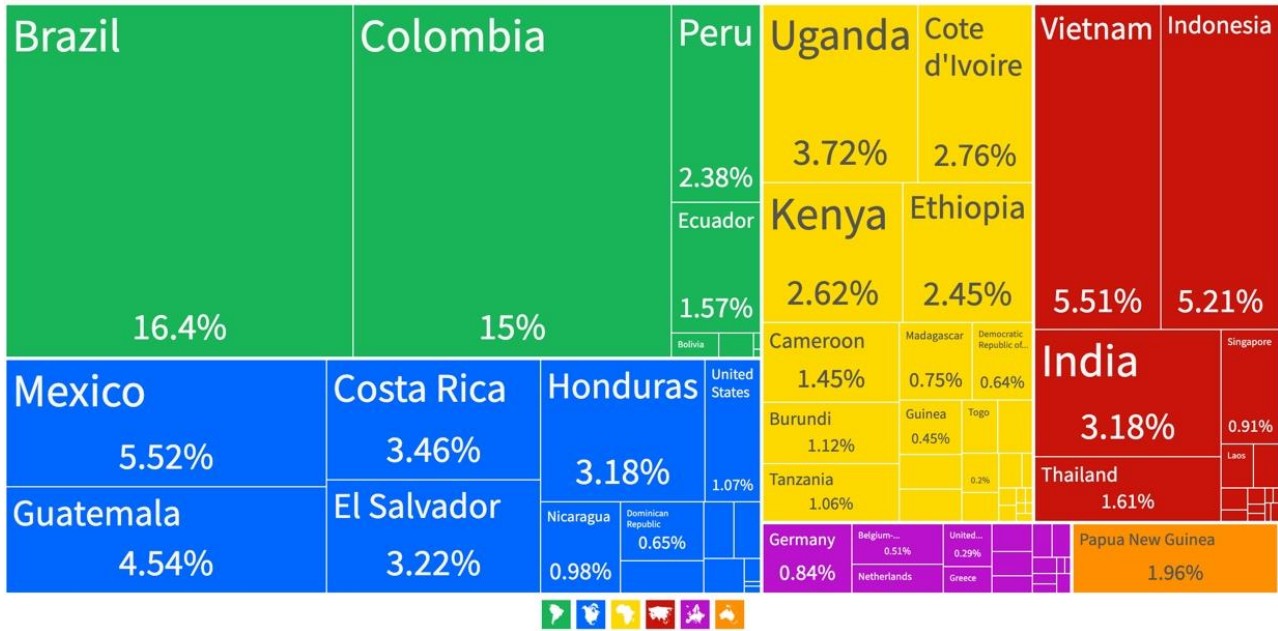

**Figure 2.** Treemap of percentage of green coffee exports by country in 1995. Different colors represent different continental zones: Green (South America), Blue (North and Central America), Red (Asia), Yellow (Africa), Purple (Europe) and Orange (Oceania).

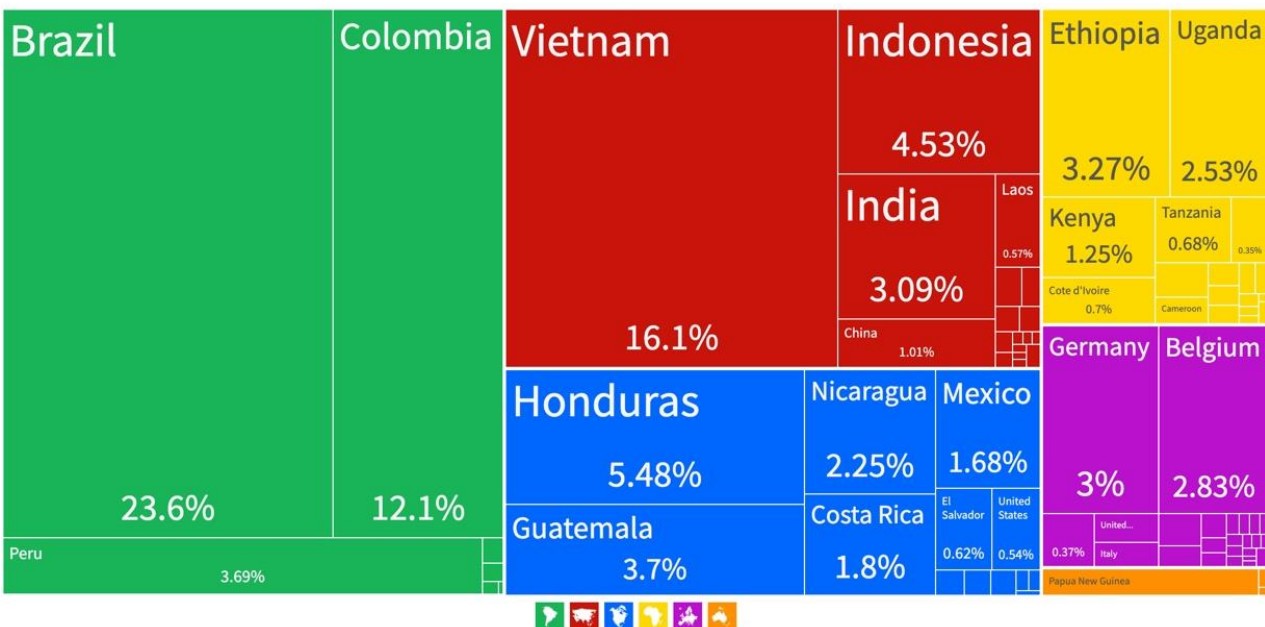

**Figure 3.** Treemap of percentage of green coffee exports by country in 2018. The colors represent different continents as in Figure 1.

*3.3. Complex Networks Analysis*

In this study, we have carried out a comprehensive analysis of the international green coffee market, using complex networks analysis, an approach used to describe real-world networks that display non-trivial interactions between components. In general, complex networks can be represented as a graph, composed by a set of elements called nodes and a set of connectors called links that represent the interactions among the nodes. Complex networks were first applied to the study of international trade using directed networks [52]. More recently, some authors have represented the international market through a complex bipartite network [53,54].

The first works that can be found in the study of the international market using complex networks are the articles by Serrano et al. [52,55]. In them, the international trade network is defined as a directed network without weight, establishing the countries as nodes of the network, and the links that unite them refer to the existence or not of a commercial relationship between the countries, that is, if there is buy/sell or not between countries. The direction of the link would indicate in which direction the net flow of the market is. The conclusions of these works focus on the topology of the network. It is observed that, taking care of the nodes, the degree distribution presents a power law, which means that there are many countries with a low degree and few countries with a high degree. Therefore, there will be a few countries that do business with many countries, while there will be others that do business with very few countries. They also found another property in the study, "assortativity", concluding that countries that have a high degree tend to join each other

Another study to highlight is the work of Fagiolo et al. [56], in which these networks were analyzed but with the introduction of weighted links with the volume of the market. Similar results were found in this work.

More recently, Wang et al. [57] used complex network analysis to find the evolution of the global coal trade, and identified that the center of the network was moved from North America to Asia.

However, if we focus on bipartite networks, such as the ones we have used in this study, Fernández et al. [53], and García-Agarra et al. [54] were the first to represent the international market through a complex bipartite network and modelized, respectively. To do this, importing countries and exporting countries are differentiated into two subgroups



of nodes, and links can only be established between countries in the different subgroups. These links are weighted and represent the sales volume of the specific product. In the work of Fernandez et al. [53], 35 products from different market areas were selected to observe the differences in market behavior between the different products.

We have used a bipartite network approach to represent the coffee trade network. For this, we have created two subgroups of nodes, one of exporting countries and the other of importing countries, and we built a complex bipartite network as follows:

1. There are two sets of nodes: exporters and importers. Links (relationships) are established between nodes of different sets.
2. A link is made between two nodes (countries) if there is a coffee trade between them.
3. The link is directed from the exporter to the importer. Additionally, the weight of the link is proportional to the volume of trade in US Dollars.

### 3.3.1. Gephi

In this research, special attention is paid to the visualization of the graphs. Therefore, we use an interactive tool, Gephi v.0.9.2, which allows us to visualize, explore and analyze all kinds of complex networks and systems, hierarchical and dynamic graphs with different designs, but also allows us to obtain quantitative measurements of the graph, which is of great importance in this study. Gephi is an interactive open-source network analysis and visualization platform written in Java on the NetBeans platform.

Through the analysis of these different types of graphs with specific characteristics depending on the parameters executed in each of them, it is possible to obtain graphs in which the international green coffee market can be seen in a clear and precise way. Each graph will have a different design and it will be possible to see the variation of the importance of each node in the network according to the parameters selected over the years, which will allow us to carry out a more exhaustive analysis of the international market. It will allow seeing at a glance the main exporting, importing and intermediary countries at a global level, as well as the importance of each node in the commercial network.

It will serve as an exploratory tool to visualize the general structure of the commercial networks, but also as a presentation tool to verify the results we have found through other methods.

To carry out this process, the necessary data have been downloaded from Un Comtrade, and after being treated and carrying out a complete cleaning of this data and applying the different parameters and algorithms available in Gephi, we obtain representative and quite intuitive images that allow a more plausible, exact and clear visualization of the international green coffee market.

### 3.3.2. Complex Networks Measures

With the graphic representation of the set of interactions, qualitative information can be obtained. Through graph theory, we can also obtain several quantitative measures that provide relative information on some aspects of the network structure. To characterize the network as completely as possible, we need a set of these measures, which can be classified into: (1) those that give local information about a node; and (2) those that give global information about the network.

Due to this, when studying the networks it must be taken into account that there are properties of the network topology that are defined for each node, thus assigning a local measure to each element of the network, while, if this information is available for all the nodes, a global property can be defined by applying some kind of synthesis operation to the values of the set of nodes, such as the maximum value, the arithmetic mean or the heavy mean. However, on many occasions, some measures, such as the mean values, do not provide any information about the system, in the same way that global measures do not inform about specific characteristics of each node. However, it is also worth mentioning that some global properties of the graphs can be obtained from the study of their decomposition into small elementary subgroups.

Therefore, depending on the type of information that is of interest, some properties or others will be chosen. The properties used for this analysis have been betweenness centrality and closeness centrality at the node level, and modularity at the global level.

## 4. Results

This section may be divided by subheadings. It should provide a concise and precise description of the experimental results, their interpretation, as well as the experimental conclusions that can be drawn.

### 4.1. Timeline Evolution of Coffee Trade

To see a correct and simple visualization of the evolution of the green coffee trade during the studied period of time (1995–2018), graphs of accumulated temporal evolution will be presented in Figure 1. These graphs have been made using a tool of The Observatory of Economic Complexity.

The trade-in green coffee is of great importance due to its large world consumption, which can be seen in its export values, which have amounted to 341.35 billion dollars US$ in the period between 1995 and 2018 [51]. As can be seen in Figure 1, the trade in coffee has varied from year to year. The peak in which green coffee exports have the lowest value corresponds to the year 2002, as opposed to the end of 2011 and beginning of 2012, which represents the highest export value of green coffee in this period of time, finding a large difference between these two years. In 2012, the European Union crisis resulted in another decrease in coffee exports. Between these two periods, there was another decline coinciding with the global crisis of 2008.

This first assessment will allow us to focus on the most important countries in our study. First of all, different treemaps of the annual green coffee exports have been generated to see the export values of each country and their evolution throughout the period studied. These treemaps show the value of green coffee exported by each country as a percentage. As an example, Figures 2 and 3 show the treemaps of green coffee exports for the years 1995 and 2018.

These figures show that the green coffee market has changed significantly. In 1995, the trade was distributed among a large number of countries, traditionally coffee producers. It was mainly concentrated in 2 countries, Colombia and Brazil, which together accounted for almost 30% of exports, while the rest of the countries, especially the majority of African countries such as Tanzania and Kenya, stood out for their low percentages. However, although they did not have as high percentages as Brazil and Colombia, in general, the North-Central and South American and Asian countries had intermediate percentage values.

In the last decade, we observe how the African countries which had a smaller export share are disappearing, ceasing to be main exporters, and how the number of exporting countries with a share higher than 1% is decreasing. Additionally, it can be seen how Central American countries are losing export share little by little, with some remaining at values very close to 1%, unlike Asian countries, which maintain a similar percentage throughout the period without losing market share, such as Indonesia. It should be noted that Vietnam, which is gradually gaining market share, has become one of the countries with the highest exports of green coffee, together with Brazil and Colombia, and has even overtaken Colombia.

Additionally noteworthy is the emergence of Germany and Belgium-Luxembourg in these main green coffee exporting countries, even though they are not coffee-producing countries, but are gaining significant market importance as exporters. Although in the first years of the study they had very little importance, they have acquired it later, even being ahead of traditional coffee-growing countries in Africa and North-Central America such as Mexico, Costa Rica, Kenya, Ivory Coast, Ethiopia, Uganda, and Costa Rica. In the case of Belgium-Luxembourg, its percentages have increased and decreased since the early

years, unlike Germany, which since it entered the 2002 coffee price crisis as one of the main exporting countries, has continued to increase its percentage of the trade.

It is worth noting that in the first years, Mexico became one of the largest coffee exporters, and yet currently has a lower export value than European countries such as those mentioned above. This is due to Mexico's decline in coffee exports and the boom in these European countries, despite Mexico being a country where coffee has traditionally been grown and produced, whereas, in the European countries mentioned above, coffee is not grown.

Brazil, as a sector policy, has increased its production to have greater participation in the market. Its productivity has increased through high-yield coffee varieties, higher intensity farming, and some mechanization of production, especially in the largest coffee-producing country [4,7].

Colombia began renovating coffee plantations at the same time that the crisis began. Through subsidies from the National Coffee Fund and direct aid from the government, the domestic price was stabilized due to the drop in the international price until January 2001. Since then, volatility, or the continuous variation in the stock market, and the drop in external prices have been assumed directly by the producer [58]. In 1996, it was displaced from second place as a world coffee producer by Vietnam [4].

It can also be seen that in the early years Africa (in yellow) and North and Central America (in blue) exceeded exports to Asia (in red), but as the years went by it overtook them with great growth due to the trade of Vietnam, leaving it only behind South America (in green).

At the end of the 20th century, the share of Arabica coffee decreased from 80% of world production in the 1960s to approximately 60%, due to the growth of Robusta coffee production in Brazil and in some African countries, as well as the emergence of the Asian continent as the main Robusta coffee-producing region in the world [59].

It should be noted that North and Central America maintained their export level at the beginning and end of the period, even though their share of trade has been reduced concerning total trade, but Africa has lost much of its export value compared to the level which began in 1995.

Central America and especially in Nicaragua suffered almost at the same time as Hurricane Mitch (in 1998) and the beginning of the coffee crisis (in 1999). When farm-gate coffee prices dropped to 30 year lows in 2001, the coffee-dependent rural economy of Nicaragua experienced, arguably, the most acute effects of the crisis in Central America [18,60].

At the end of the 1990s and the beginning of the 2000s, in Central America (Arabica producers), the production costs were very high and could not be covered with the prices obtained. The abandonment of coffee plantations or the elimination of care, such as fertilization, reduced the harvest [58]. In 2000, the Nicaraguan coffee sector generated US$171 million in export earnings, 26% of all export revenues in Nicaragua [60]. In 2021, Nicaragua's coffee exports dropped by 50% and an estimated 500–3000 coffee farms succumbed to foreclosure [61]. In response, banks cut their lending by 80% from 1999 to 2001. Three private banks financing coffee production declared bankruptcy and rural workers and peasant farmers who depended on coffee wages or sales as the main source of household income suffered rampant unemployment, farm foreclosures, and food shortages [60,61].

Unlike the other continents, it can be seen (Figure 1) that in the period 2011–2012, there is a high peak which is out of line with the rest of the graph because such high values had never been reached before; in Africa, the value for the period 2011–2012 is not so far from that of 1995. This means that the market in Africa has not undergone as many variations as in the rest of the continents but has suffered a great decrease in the value of exports over the years. The countries with the highest exports are Ethiopia, which has managed to increase its exports, and Uganda, which has managed to maintain its exports. In global markets, Ethiopian coffee is valuable because it is of the Arabica type and because of its unique taste [62].

Finally, Europe has considerably increased its level of green coffee exports despite not being a coffee-growing area.

*4.2. Coffee Trade Complex Networks*

Through complex network analysis, different characteristic properties of the graphs have been studied to see the evolution of the international coffee trade over the years. In this way, global knowledge of how the international market is distributed can be obtained, as network analysis allows a global vision of the interactions.

4.2.1. Grade Distribution

The degree of a node is the number of links that that node has, so we will know if this node is very connected or not connected at all.

A node with many links ("hub") will be an important node, as it is related to other nodes.

As the networks in the study are networks that are directed because they deal with exports of green coffee from one country to another, we will talk about output degree, input degree, and total degree.

Figure 4 represents the input degree of different countries into the international green coffee trade in 2002. The countries with the highest input degree are Belgium, the United States, the United Kingdom, Japan, and Denmark, as they are the countries with the most entry links.

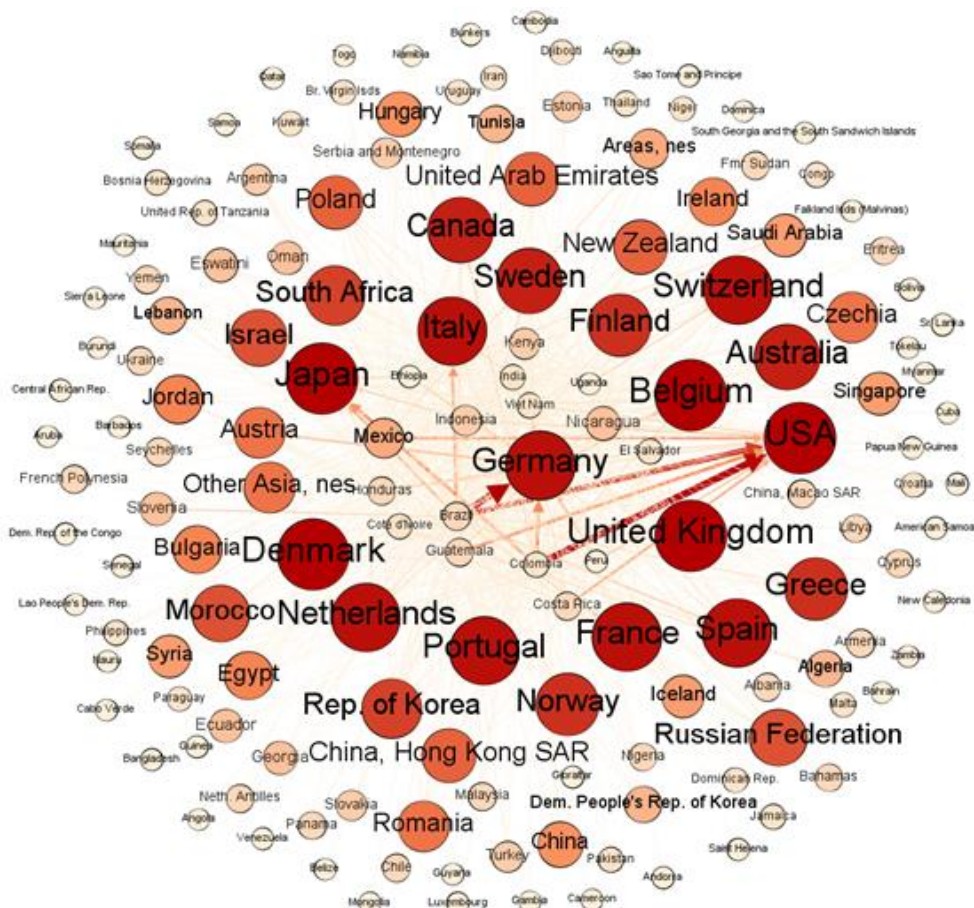

**Figure 4.** Input degree of green coffee exports in 2002. The size and color of the node indicate the input degree. The larger the node size and the darker the red color, the higher the input degree.

Figure 5 shows the output degree of countries from the international green coffee trade in 2002. The countries with the highest output degree, and therefore with the most outbound links, are Vietnam, Indonesia, India, and Brazil.

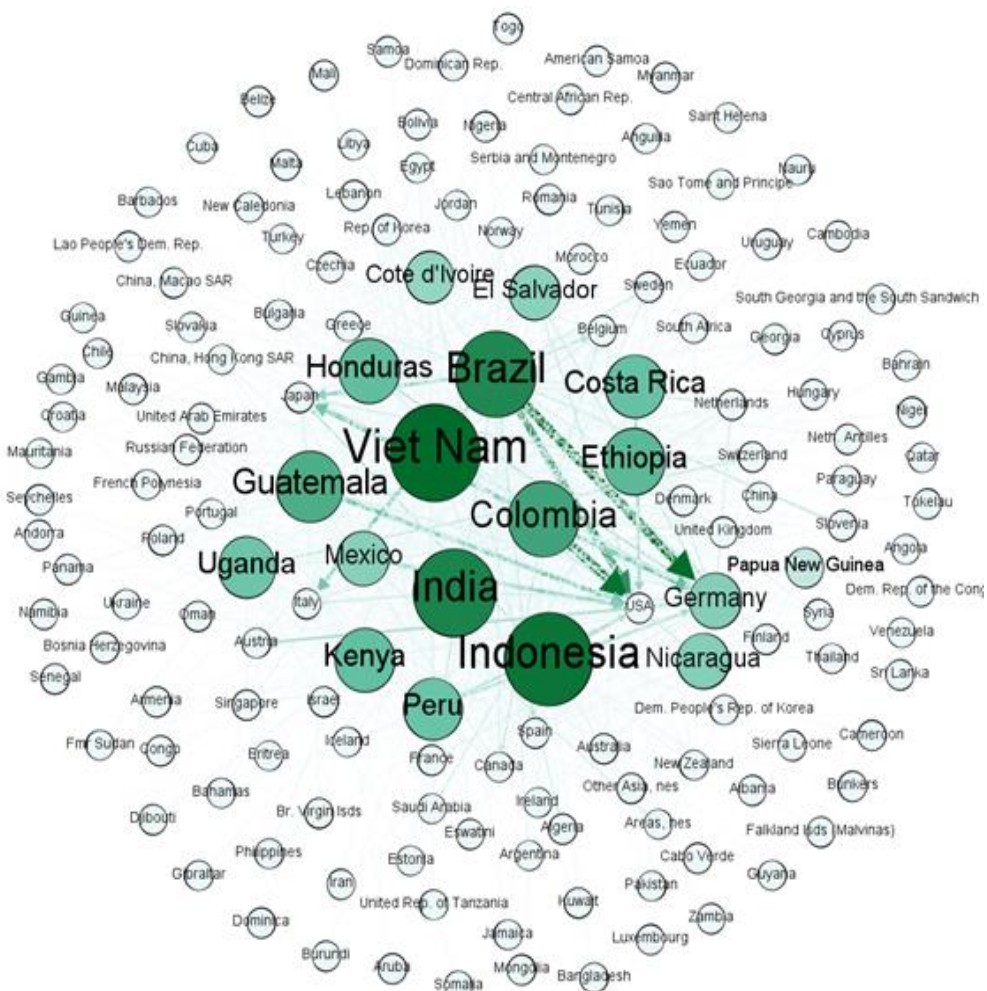

**Figure 5.** Output degree of green coffee exports in 2002. The size and color of the node indicate the degree of output. The larger the node size and the darker the green color, the greater the degree of output.

However, working with weighted complex networks, degree with weights, weighted output degree, and weighted input degree can also be considered, as when taking weight into account, the results obtained in terms of the importance of countries in the international green coffee trade are quite different from those obtained without taking weight into account. To compare this, the complex networks will be presented as weighted networks.

Figure 6 shows the complex network of international trade in green coffee according to the weighted input degree of the different countries in 2002. As can be seen, the countries with the highest input degree with weights are the United States and Germany. When compared to Figure 4, it can be seen that in Figure 6, there are only two large countries, while without considering the weights, the number of countries with relevant importance increases. In Figures 4 and 6, the United States is one of the most relevant countries and is, therefore, a very important importer. However, although Germany is relevant with regard to its degree of entry taking into account the weight, or not, it acquires greater importance with regard to the degree of entry with weights.

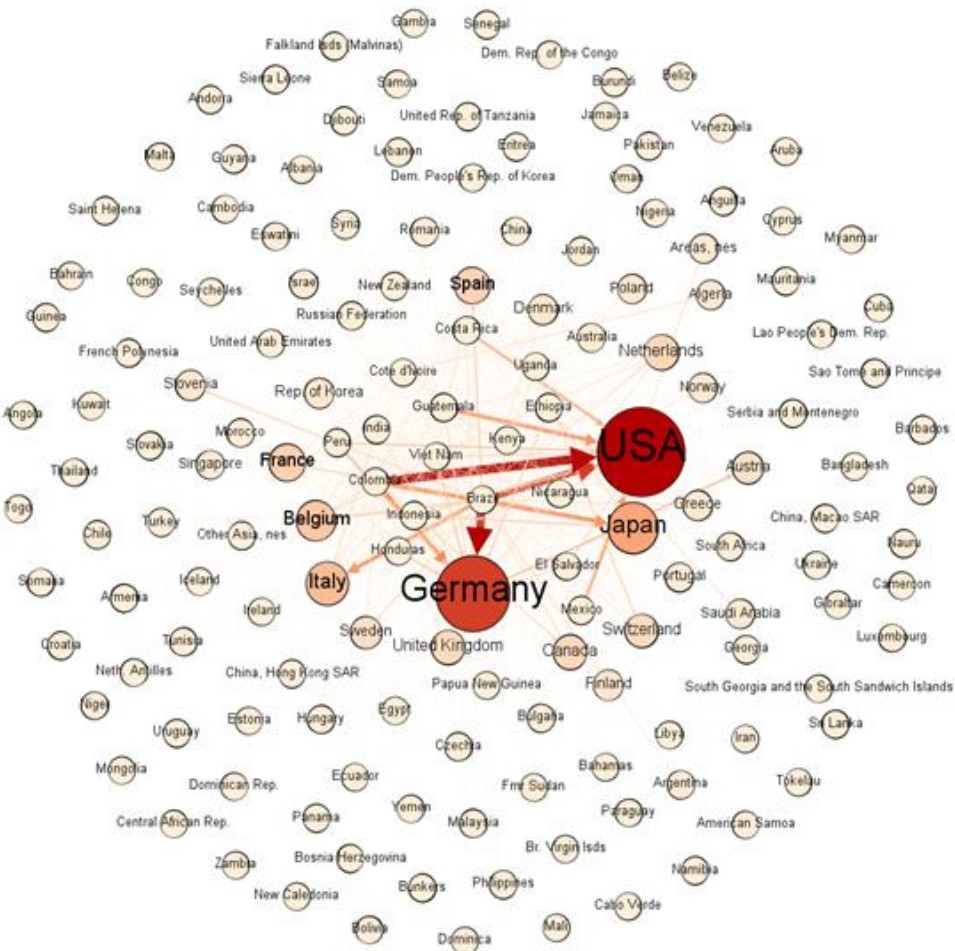

**Figure 6.** Input degree with green coffee export weights, 2002. The size and color of the node indicate the input degree with weights. The larger the node size and the darker the red color, the greater the weighted input degree.

Figure 7 represents the output degree with weights of countries in the international trade of green coffee in 2002. Colombia and Brazil stand out as major exporters, as they are the ones with the greatest output weight. In the same way that it happens when comparing the input degrees with and without weight, when comparing Figure 5 of output degree with Figure 7 of the weighted output degree, it can be seen that in the case of Figure 5 several countries stand out, while in Figure 7 only Colombia and Brazil have great relevance. It can be seen that countries such as Vietnam, Indonesia, and India are the countries with the highest degree of exit because they have many connections with other countries, but do not have a high weighted output degree. Unlike Colombia, which stands out more for its degree of exit with weight than without. Brazil is of great importance due to the output degree both with and without weight but is much more important due to its weight because of its high volume of exports.

However, to define a network in more detail using a global magnitude according to the degree of the nodes, the degree distribution is analyzed.

Firstly, the evolution of degree distribution has been analyzed using bar charts over different years of the period under study (1995–2018). Figures 8 and 9 represent the degree distribution of the network over several years:



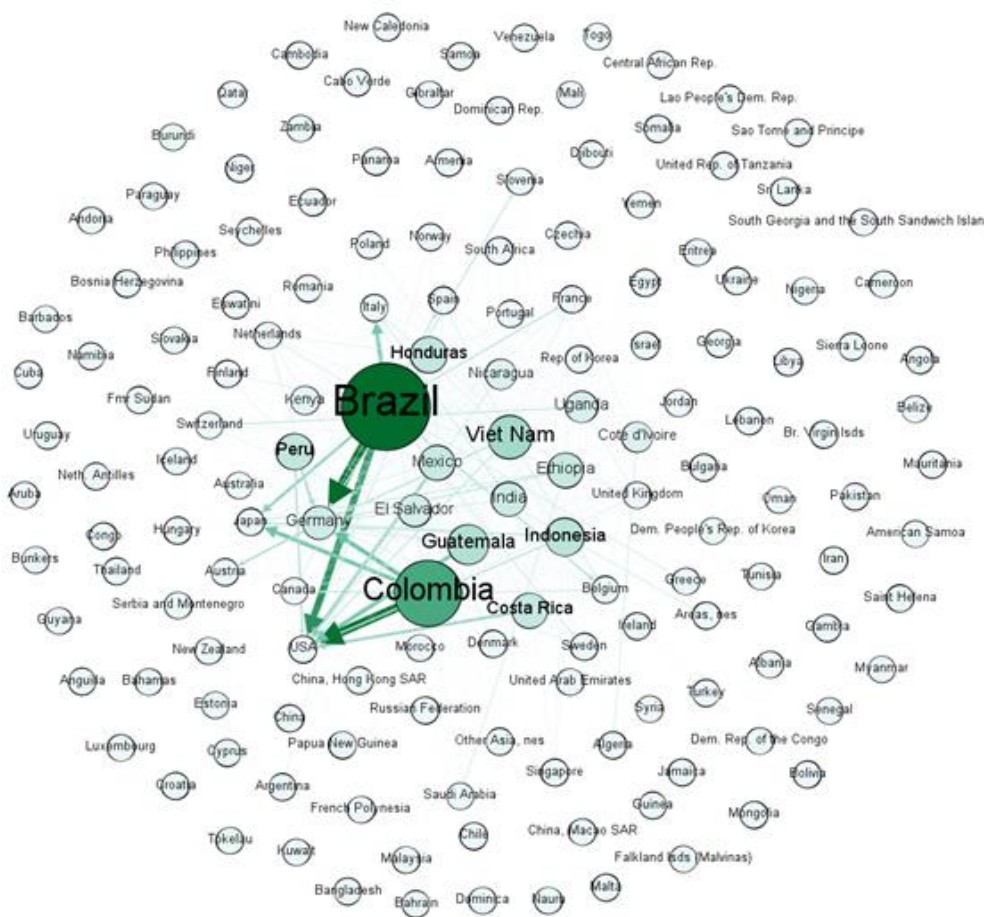

**Figure 7.** Output degree with weights of green coffee exports in 2002. The size and color of the node indicate the output degree with weights. The larger the node and the darker the green color, the higher the weighted output.

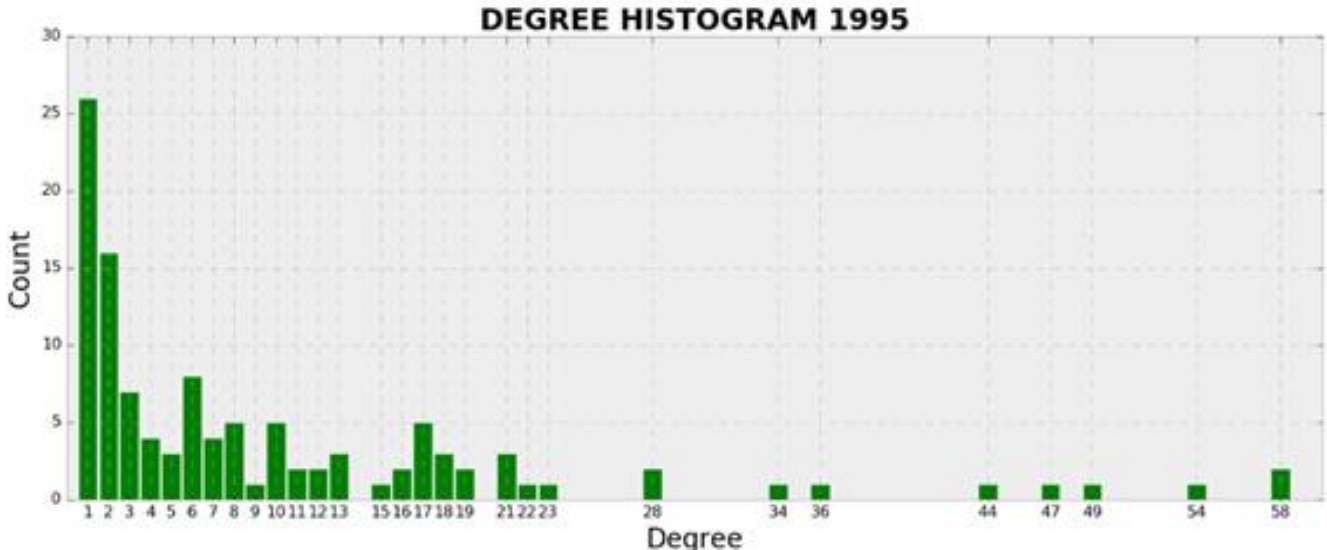

**Figure 8.** Graph of degree distribution in 1995.

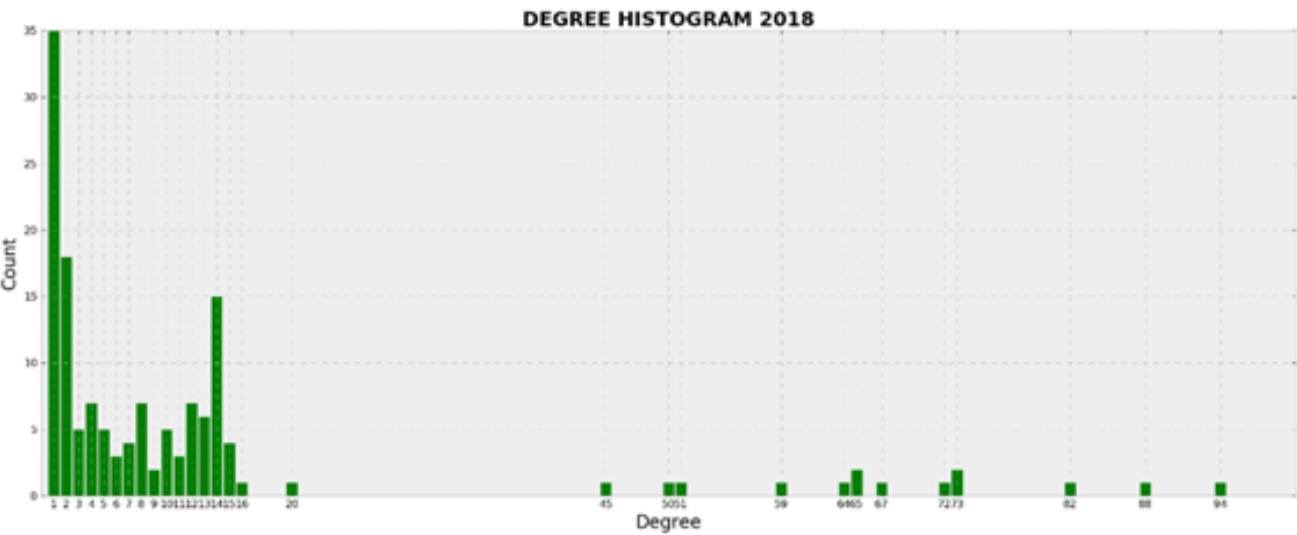

**Figure 9.** Graph of degree distribution in 2018.

After analyzing the graphs for the different years (Figures 8 and 9), it can be seen that they all have a similar shape, with many countries trading with only one country, and few countries trading with a high degree, meaning only one or two countries are trading with most countries, being the main importers or exporters. It can therefore be concluded that these networks correspond to the so-called "Free Scale" networks and that the international coffee market is heterogeneously distributed.

Therefore, it can be concluded that in the international coffee trade, there are a few countries that are the main traders, which have specialized in importing or exporting over the years, As a result, at the beginning of the period under study, trade was more evenly spread across countries, but as the years have passed, the large exporting and importing countries have become stronger, making the market almost exclusive to them.

These large exporters, which account for almost all the trade, are the main coffee-producing countries such as Brazil, Vietnam, and Colombia. While the big importers are countries that consume a lot of coffee like the United States, or countries like Germany and Belgium-Luxembourg that import a lot of coffee, but not only to consume it but also to sell it, acting as intermediaries. As a consequence of this evolution, small coffee-producing countries lose weight and prominence in the green coffee bean market, placing themselves at a negotiating disadvantage, which may compromise the sustainability of coffee production or the incorporation of new coffee growers to the production system.

4.2.2. Betweenness Centrality

This property will indicate the importance of a node in the network depending on how many times the node is an intermediate step between the shortest path of two nodes [63].

Figures 10 and 11 show the betweenness centrality of the complex network of international trade in green coffee over several years.

When looking at how the international green coffee trade network has evolved, it can be seen that in the first years, the importance of countries in terms of intermediation was spread over a greater number of countries, being these the traditional producers of green coffee, mainly developing countries in Latin America. However, as the years have gone by, this importance has been concentrated in a smaller number of countries, among which some are coffee producers, but others are not.

Figures 10 and 11 show the incorporation, between 2002 and 2017, and the importance of non-coffee-producing countries as intermediaries in the global green coffee trade network to the detriment of Latin American producing countries. Germany and Belgium with a high value of betweenness centrality have become important intermediaries

in the global green coffee trade network, making them even more important than the coffee-producing countries.

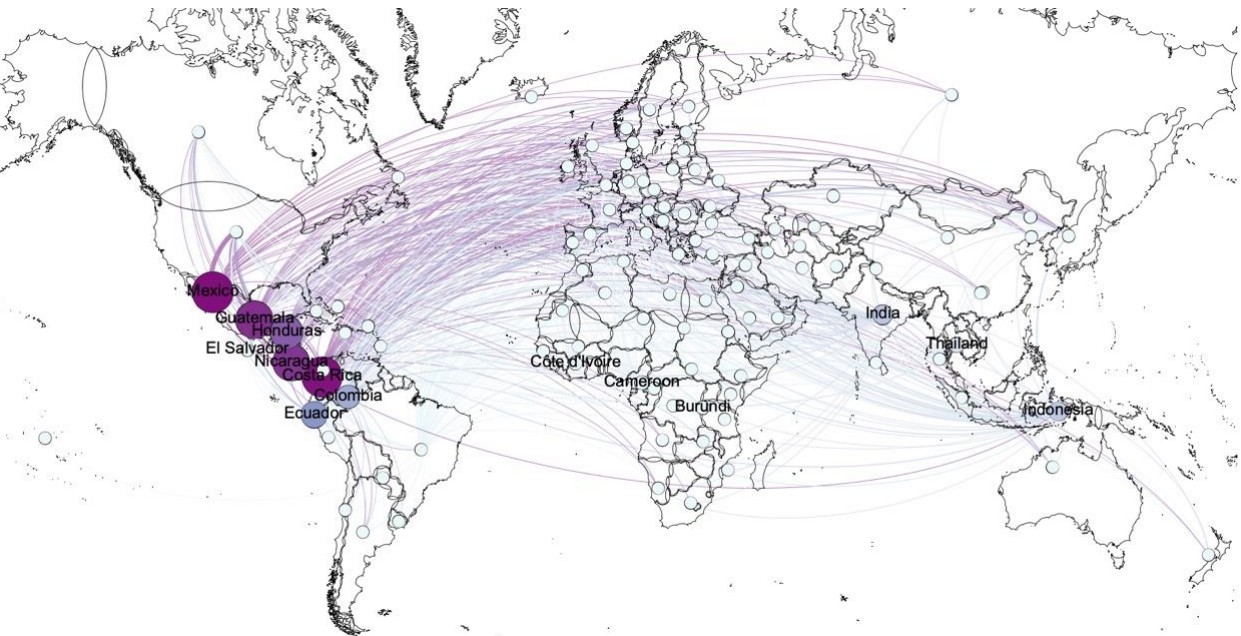

**Figure 10.** Betweenness centrality of the complex network of international trade in green coffee in 1995.

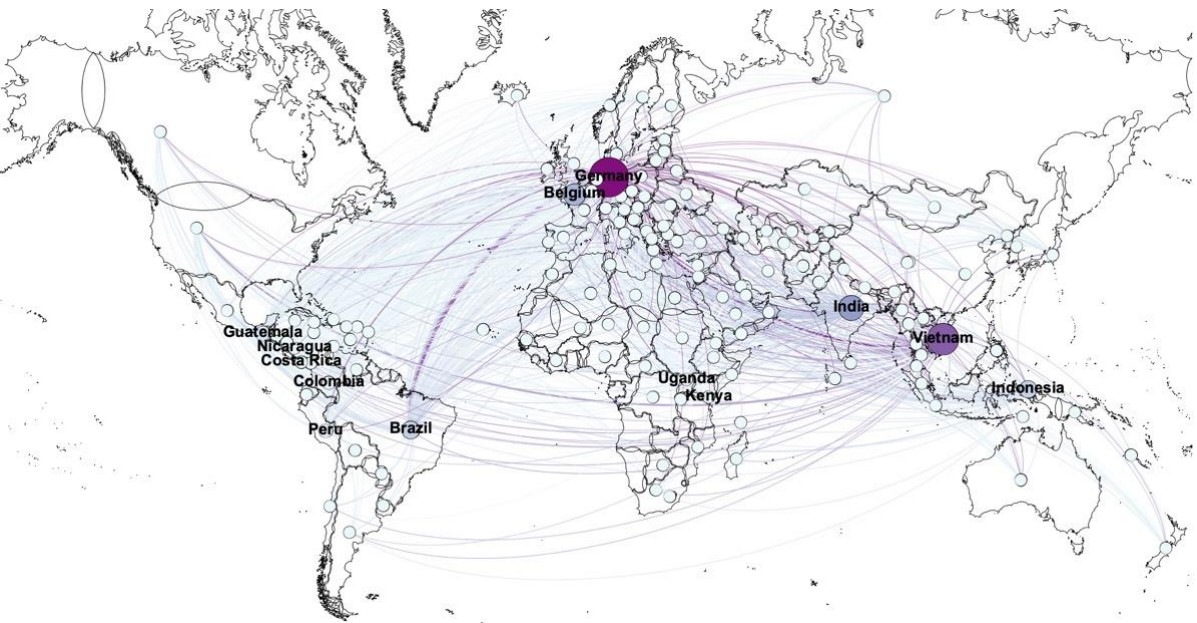

**Figure 11.** Betweenness centrality of the complex network of international trade in green coffee in 2017.

In 1975, the commercial exchange between the producing countries made the centrality of intermediation high among them, for the simple fact of being sellers and buyers with a good volume of market. But when the international market has developed, true intermediaries have appeared, that is, countries that serve as buyers and sellers, without being true coffee producers.

### 4.2.3. Closeness Centrality

Another property of centrality that will be studied is the Closeness centrality, which will indicate the importance of a node according to the proximity of other nodes [64].

Betweenness and Closeness centralities are both centrality measures using the shortest path between nodes. Betweenness centrality is generally regarded as a measure of others' dependence on a node, and therefore as a measure of potential control. Closeness centrality is usually interpreted either as a measure of access efficiency or of independence from potential control by intermediaries. But both measures are topological measures, not geographical measures.

Figures 12 and 13 show the complex networks of international trade in green coffee as a function of their closeness centrality.

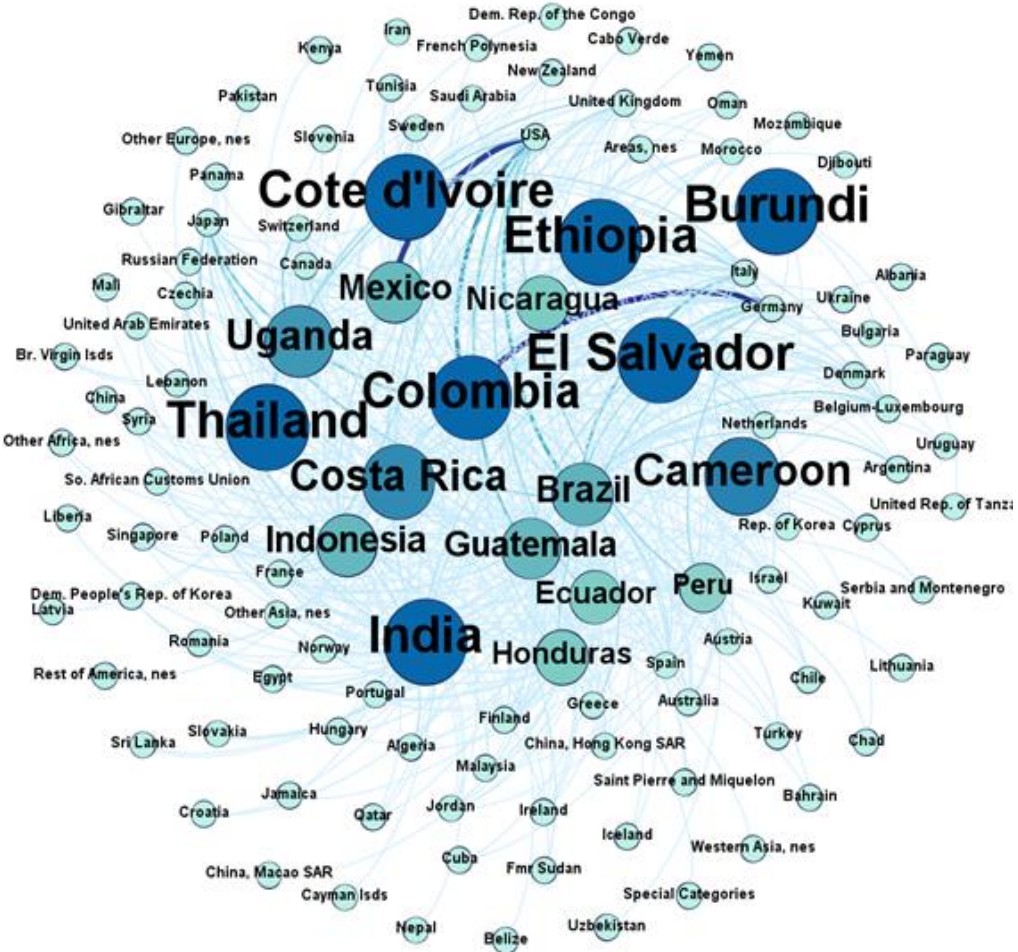

**Figure 12.** Closeness centrality of the complex network of international trade in green coffee in 1995. The size and color of the node indicate the closeness centrality. The larger the node and the darker the blue color, the higher the weighted output.

Therefore, once this property has been analyzed, we can say that, according to these criteria, the importance of countries is not related to the volume of exports they carry out, since there are many very important countries in the network due to their Closeness Centrality value despite their low export volume. What makes these countries important is the topological distance they have from other countries. Therefore, even if a country is small and has a low degree value or low export volume, it can be very important if it is connected to highly connected nodes since it is positioned at the center of the network, as is the case with some countries such as the Ivory Coast in 1999 and Papua New Guinea in 2000. For these small- or low-volume producing countries, identifying the important

nodes of the network to which they are connected can be useful in defining trade policies. But it can be seen how, as the years go by, the main exporting countries such as Brazil and Vietnam are becoming more important than the small coffee-producing countries, as well as non-coffee-producing European countries such as Germany. In other words, the links that existed in 2002 between small producers and large, well-connected nodes in the network are gradually diminishing.

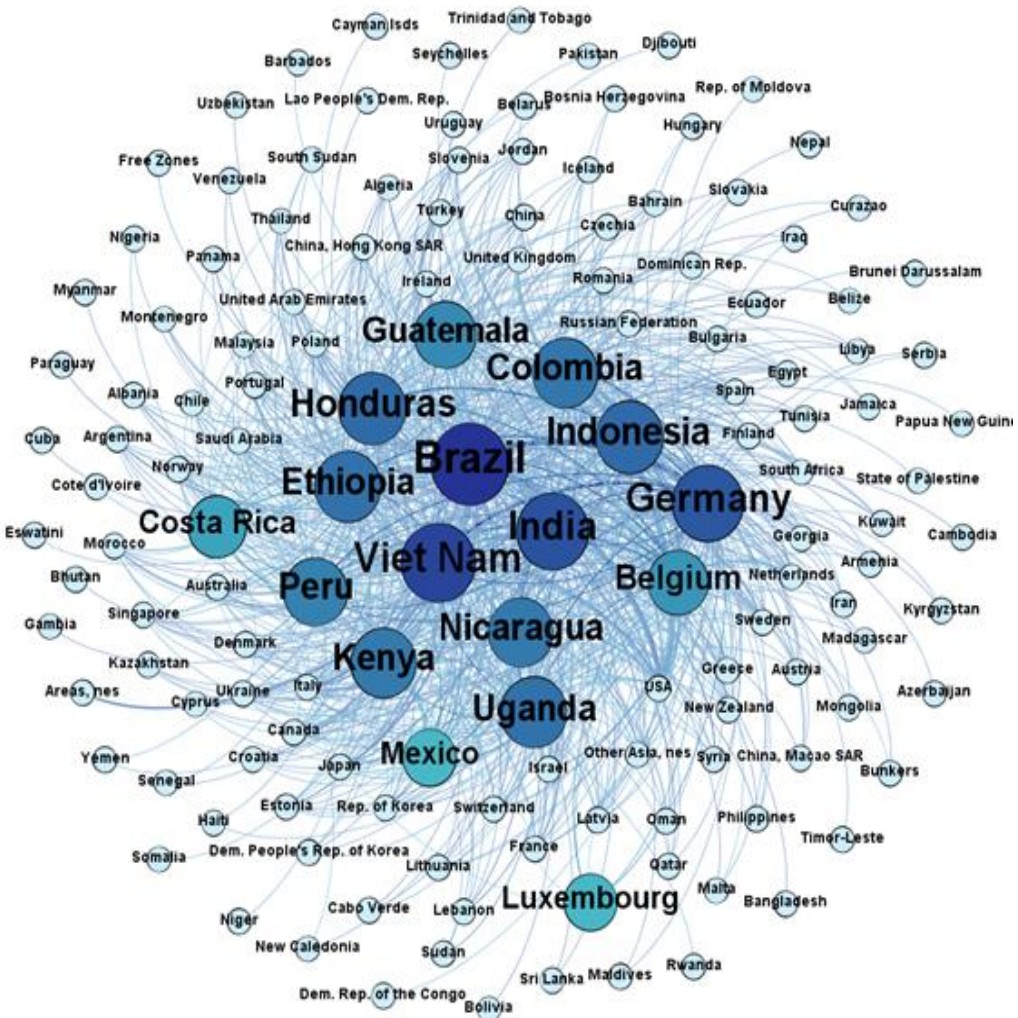

**Figure 13.** Closeness centrality of the complex network of international trade in green coffee in 2017. The size and color of the node indicate the closeness centrality. The larger the node and the darker the blue color, the higher the weighted output.

Given the value of closeness centrality, it can be seen in 2002 how the traditional coffee-growing countries in Africa have high importance, unlike other properties that do not give these countries any importance in the coffee market network. However, in 2017, countries such as Cote d'Ivoire and Cameroon have completely lost connection with the central nodes of the network; something similar happens with El Salvador. However, countries such as Germany, Belgium or Luxembourg emerge as nodes of the network with high Closeness Centrality value.

### 4.2.4. Modularity

Finally, it will be analyzed how the nodes of the network are grouped. With this property, it is possible to obtain the different groups of countries that are formed according to the relationships that are established between them. Those that have similar characteristics or specific relations and are connected between them will form a group, a community,

while others that do not meet these characteristics will not be part of that community but will be isolated or grouped in another community.

Below are the commercial networks of several years divided into the communities that Gephi has established.

The communities will be represented by different colors, also showing the percentage of nodes that are part of each community

Firstly, modularity by communities and colors of the international green coffee trade network in 1995 is represented in Figures 14 and 15.

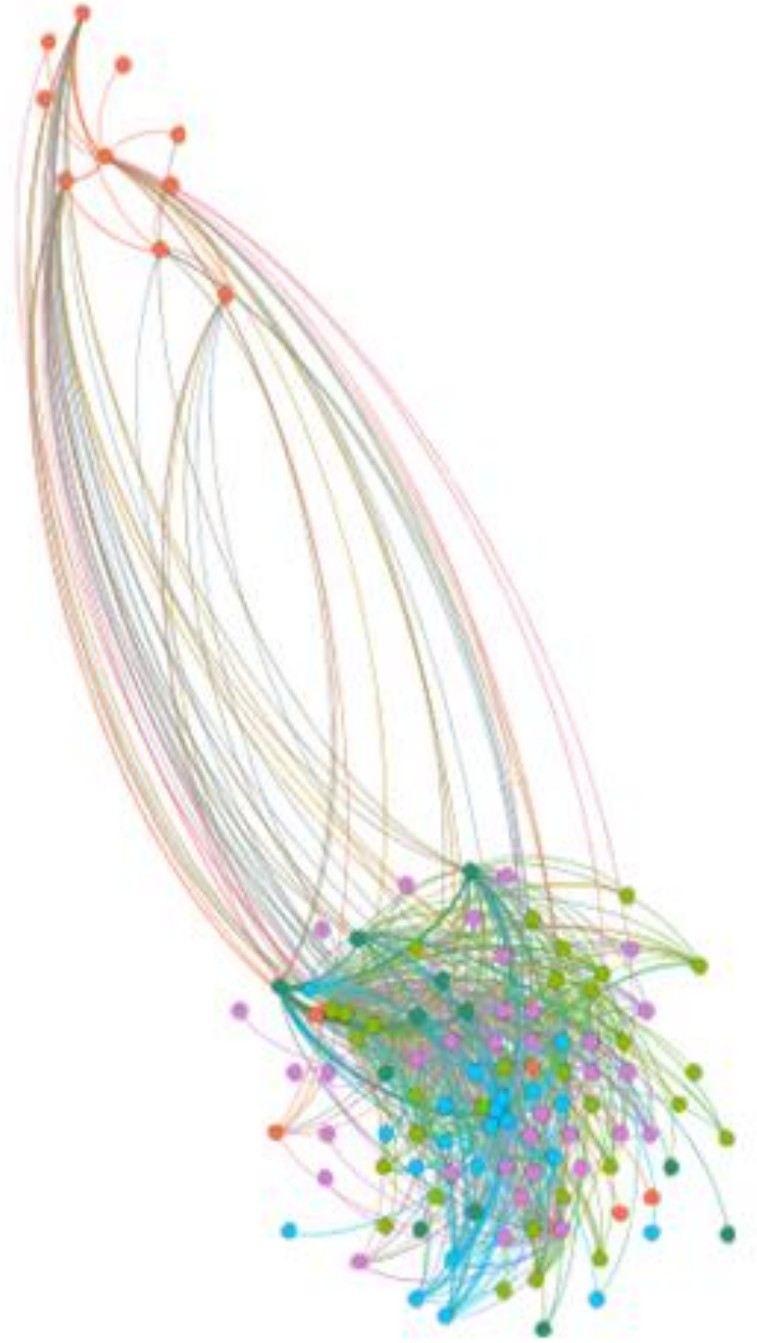

**Figure 14.** Modularity of the complex network of international green coffee trade in 1995. Colors indicate the modularity class (see code in upper left panel in Figure 15).

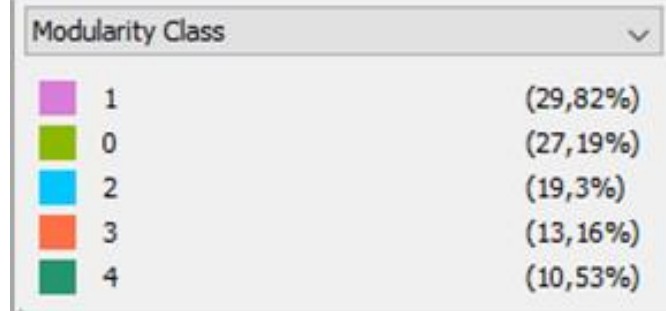

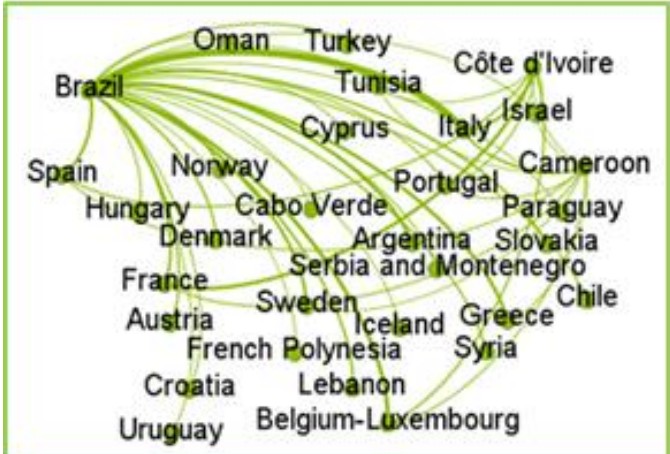

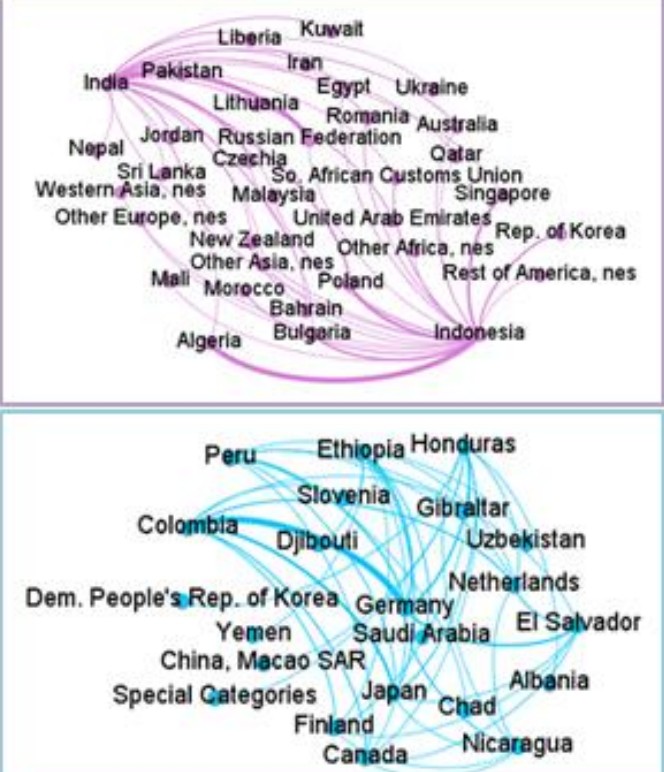

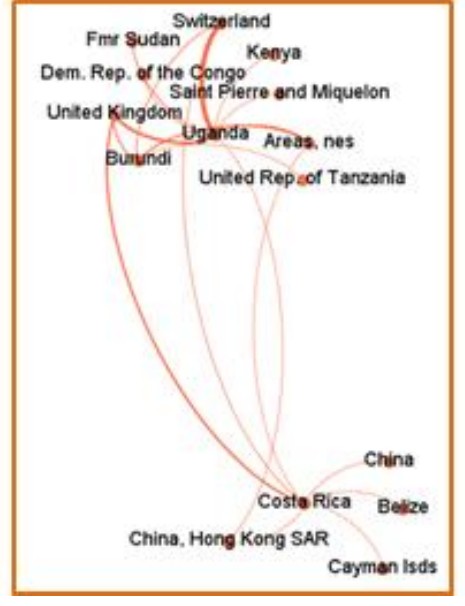

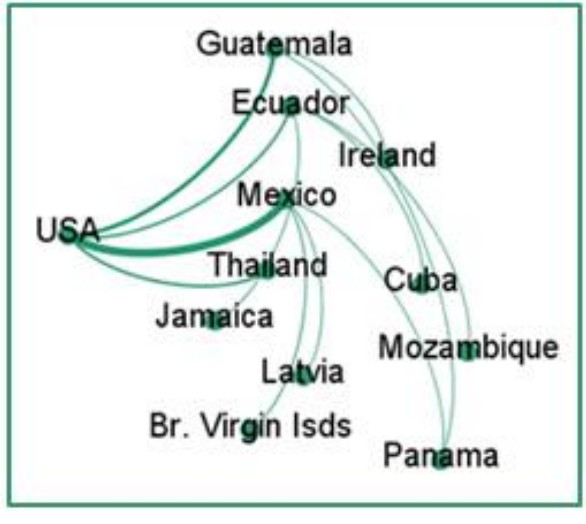

**Figure 15.** Modularity by communities and colors of Figure 14 of the international green coffee trade network in 1995.

Below, modularity by communities and colors of the international green coffee trade network in 2017 are represented in Figures 16 and 17.

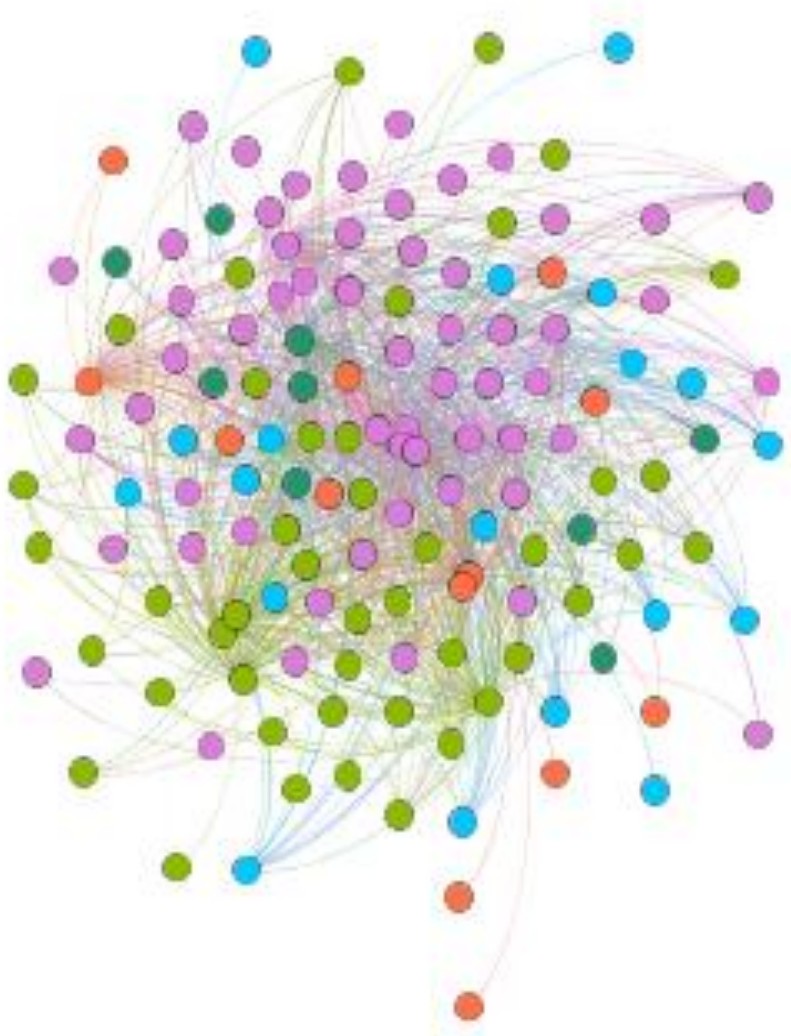

**Figure 16.** Modularity of the complex network of international green coffee trade in 2017. Colors indicate the modularity class (see code in upper left panel in Figure 17).

Once the modularity of different years of the period under study has been analyzed, a certain tendency of groupings according to exporting and importing countries can be observed. It can be said that the communities are divided into major exporting countries and corresponding importing countries.

It can be seen that, in general, one of these communities is made up of large Asian exporters such as Vietnam, India, and Indonesia. Another community usually consists of Brazil as a major exporter and Germany as a major importer. Additionally, there is usually another community made up of the exporting countries of Central America, with the United States as a major importer of these. As for the smaller coffee-exporting countries in Africa, they are often grouped in these large communities of other major exporting countries. It can also be seen how in many cases these communities tend to be formed by attending to the different continents.

But it can be seen that, unlike in the early years, in the latter part of the period, there is a tendency to form smaller communities, with a lower percentage of nodes, in which a small exporting country in Africa is normally found with its importers. Besides, some medium-sized communities are also seen to be formed by an exporting country from Asia.

It is difficult to understand the structural change of modularity over time. The modification of the coffee market shown may be due to different factors—macroeconomic, geopolitical, etc.—which require different and extensive specific studies on each aspect that exceed the objectives of this article.

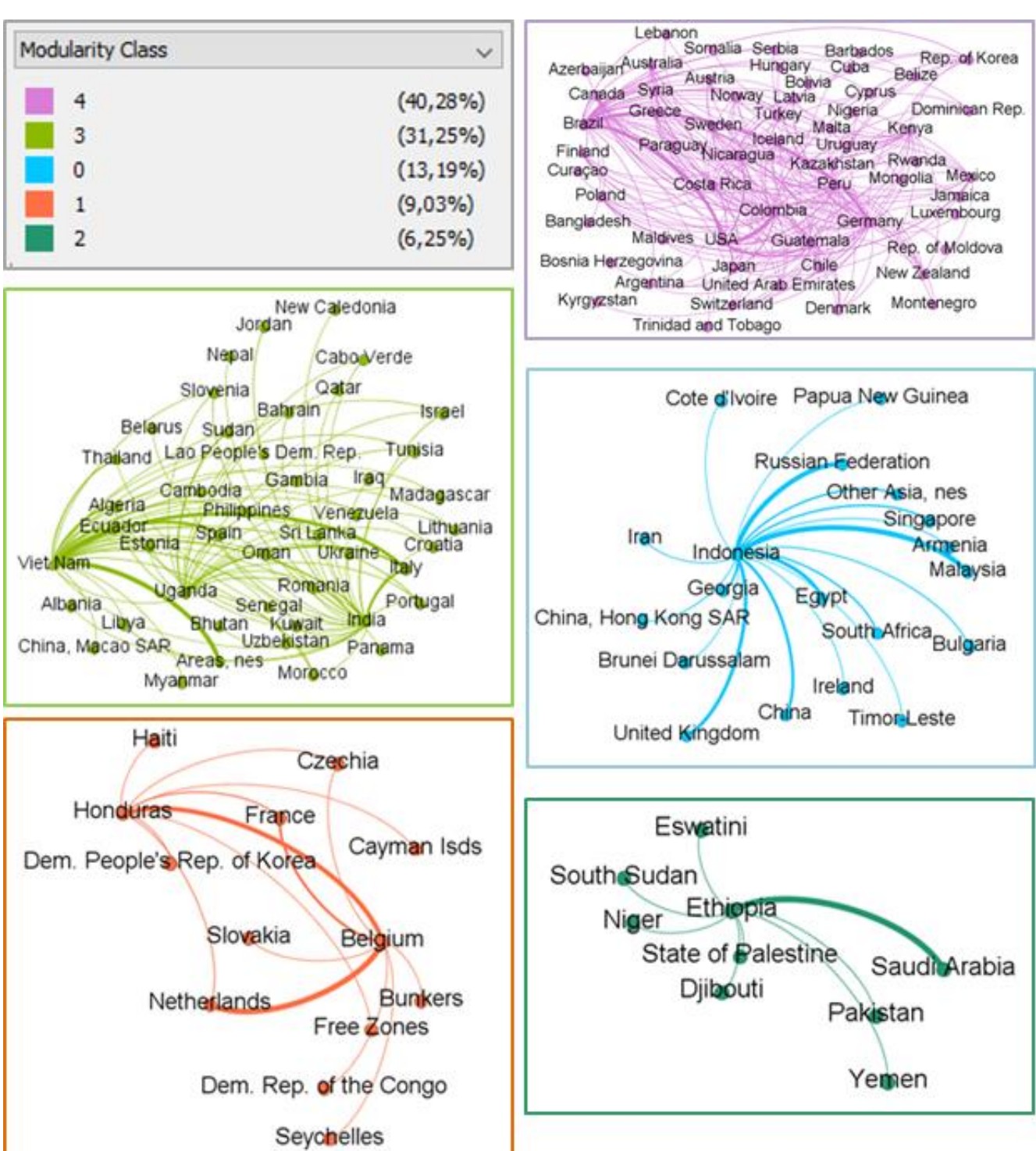

**Figure 17.** Modularity by communities and colors of Figure 14 of the international green coffee trade network in 2017.

## 5. Conclusions

After carrying out the study of the international green coffee trade over a period of years, using modern data analysis techniques, we can make several conclusions.

On the one hand, concerning the data analysis techniques used, such as the construction and visualization of graphs and data mining, it has been proved that they can be a useful element in exploring, analyzing, and understanding the large volume of data

available on the complex international green coffee market in a clearer, simpler and more effective way. They allow us to determine what is happening in the international market and what role the different countries play globally, being able to visualize at once the world market in a complete way. Through tools such as complex networks, it is possible to understand how green coffee transactions take place between different countries in the world market reliably and more simply, in addition to seeing through their visualization many characteristics of the international green coffee market that only with the data would not be seen and would go unnoticed, such as that a country that does not grow coffee like Germany is a great marketer of this product. This type of analysis has already been carried out comparing different markets, for example by Basevi and Biggiero [50].

With this analysis, new ways to diagnose a country's problems and thus to develop a political and economic strategy emerge. These new policies would contribute to improve the sustainability of coffee production chains, thereby alleviating the precarious situation of coffee-producing countries. Complex network analysis could also be used to study what would happen in the international market if we were to make certain countries disappear, such as one of the big exporters or importers of green coffee, so that it would be seen how this would affect commercial transactions and the rest of the countries.

By forming this market, such a connected network implies different situations because while things are going more or less well, no problems arise because the network is very robust, but when it goes wrong, the problems immediately pass to many of the actors involved in the network that are connected.

This study has also highlighted the importance and the socio-economic reality of the agri-food chain, a system that brings together both social and economic actors who are interrelated and carry out activities that add value to the product studied, from its production to its arrival at the consumers. This chain should seek greater transparency in business transactions and balanced and sustainable work and benefits among the actors involved in the different processes.

For this reason, this research verifies that an exhaustive study of a certain agricultural product can be carried out through a good and updated database, combined with an adequate treatment of these data through different computer applications.

Through these studies, it is possible to see how digitalization, data mining, and interactive data visualization tools contribute to socio-economic progress and have a direct application in the development and wealth of countries.

On the other hand, after analyzing the international green coffee market through "the observatory of economic complexity" [50] it is possible to conclude that the market volume during the period studied has been constantly increasing and decreasing, finding a minimum in 2002 and a maximum in 2012, suffering a considerable general increase from the first years of the period until recent years. Throughout these years it has been observed how the importance of the different countries in the market has varied, as in the first years, the market was spread over a greater number of countries than in recent years. Furthermore, in the beginning, the market was divided between the traditional coffee-growing countries of South and North-Central America mainly and also Africa, while as the period studied progressed, the countries of North-Central America and Africa became less important and moved to countries in Asia and also in Europe, among which Germany and Belgium stand out. Despite their importance in the trade, these countries do not grow coffee, which means that these countries, which act as intermediaries, acquire great importance, even more so than other traditional coffee-growing countries in Central America and Africa. It is also worth noting how the large producing countries (Brazil, Colombia and Vietnam) are increasingly taking over trade, making small exporters less important. Brazil stands out as the largest producer, often accounting for one-third of total world coffee exports. In the case of Vietnam, the importance it has acquired over the years stands out, being a country with a low percentage of exports at the beginning of the period, but one of the most important in recent years. While these countries focus on mass production without regard to quality, other countries have smaller but higher quality

production. Looking at all world exports, the global market is based mainly on exports from South America, due to the production of Brazil and Colombia.

Through the grade distribution, we have been able to verify how the countries with the greatest prominence are those that are only exporters since their entry grade is zero. We have also been able to see that at the beginning of the period the market was distributed among more countries than at the end of the period, which means that the large exporters and importers have been specializing and covering the whole market, causing the smaller countries to reduce their market and lose importance.

Looking at the betweenness centrality, also at the beginning of the period, a greater number of important countries for intermediation were found, being located mainly in the Central American countries, while at the end of the period, the number of relevant countries decreased, being the main intermediaries of the international green coffee trade network countries of the European market such as Germany and Belgium, and also Asian countries such as Vietnam and Indonesia.

Regarding the analysis through closeness centrality, even though greater importance is given to small coffee-producing countries in Africa than with other properties studied, a clear trend can also be seen in the way Brazil, Vietnam, and Germany are gaining importance.

Finally, looking at the modularity, there is a trend of grouping the main producing and importing countries, generally finding only one large exporter in each group along with other small exporters, or on the contrary, several exporting countries, generally from the same continent.

So, this research can conclude that the international market for green coffee has varied considerably over the period studied. It has gone from being distributed among the traditionally coffee-growing countries of Africa, South America, and North-Central America, to being distributed among the three main coffee-producing countries, with different Asian countries producing and acting as intermediaries acquiring great importance, as well as European market countries that act exclusively as intermediaries, since coffee is not grown in them.

This study also shows how, despite the increase in consumption and production of coffee, producing countries are losing power to importing countries, causing wealth and added value to remain in developed countries, and therefore inequalities between developed, and developing countries continue to increase. Even though the world market for green coffee is growing and generating more economic value, this is not reflected in the producing countries, as most of the wealth it provides remains in the importing countries and not in the producers. This fact can generate sustainability problems in small producing countries since they are losing global market volume, although within countries this market is very important in their gross domestic product.

This is why countries that depend so much on coffee production should be urged to improve the marketing of the product, trying to add value in the producing countries in order to increase the GDP and obtain greater wealth and reduce poverty in these areas, as well as favoring local trade, avoiding so many intermediaries. In this way, a greater percentage of profits would remain with the producers and not the majority in the rest of the value chain, as is currently the case. It is also necessary to raise awareness of these differences between the different actors in the value chain so that they disappear and promote fair and equitable trade that does not produce so many inequalities.

**Author Contributions:** Conceptualization, R.R.-R., V.N., V.D.-B., M.B. and J.G.; methodology, R.U.-C. and J.G.; software, R.U.-C. and J.G.; formal analysis, all authors; data curation, R.U.-C. and V.N.; writing—original draft preparation, R.U.-C., J.G. and R.R.-R.; writing—review and editing, R.R.-R., V.D.-B., M.B. and J.G.; visualization, R.U.-C. and J.G. All authors have read and agreed to the published version of the manuscript.

**Funding:** This research was supported by the Ministry of Science, Innovation and Universities-Spain under Grant No. PGC2018- 093854-B-I00.

**Institutional Review Board Statement:** Not applicable.

**Informed Consent Statement:** Not applicable.

**Data Availability Statement:** Data are public data. They can be obtained in [49].

**Acknowledgments:** The authors would like to thank to the Plataforma de América Latina of the Universidad Politécnica de Madrid for their support and funding of this publication.

**Conflicts of Interest:** The authors declare no conflict of interest.

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
