# Peer review of "Growing Inequality in the Coffee Global Value Chain: A Complex Network Assessment"

_sustainability, doi:10.3390/su14020672_

Round 1

Reviewer 1 Report

You are using global coffee trade patterns in an attempt to claim growing inequality in the global value chain, but besides the fact that some exporters have grown more dominant and gained greater market shares over a 20-year period, you are not demonstrating "inequality" in a moral or normative sense. Quantitatively you are showing trends, but if your intention is to highlight economic "inequalities", you have not shown how these trends have hurt certain exporting countries or contributed to downward pressures on prices.

In my view, what you have done is taken a dataset of trade figures (imports and exports of green coffee beans, presumably in dollars and not volume) and run these through a network analysis application. Some of the resulting graphs are instructive, while others are quite confusing and don't seem to add much to understanding. It is a bit unclear what your database consists of (which fields it contains), but if it is just exports and imports of a certain HS code, running this through an advanced network program seems like an overkill. However, if your data contains traceable data on intermediate transactions (e.g. beans sold and transiting through intermediate countries), a network analysis would be justified. The nature of your data is not clear, so this should be explained better. 

What is frustrating for a geographically inclined reader such as me is how you have presented figures 4-7, 12 and 13. I feel all these circles should be implanted on a world map instead of appearing randomly in big spheres. Would not a combined version of figures 10 and 11 tell us all we need to know? Here we can actually see trade relationships and magnitudes. I'm not sure how to interpret "betweenness centrality" as long as you have not indicated whether some of the coffee transits between several countries (nodes) before it arrives at the destination. It is mystifying why in 1995 the Central American countries have high betweenness scores, as you would think they would be direct exporters? I can sort of understand the high Belgium/Germany betweenness scores in 2017, even though you don't explain why these two countries have become such important intermediaries. In the same vein, does your data reflect monetary flows or the actual physical flows of the coffee beans? In other words, are coffee contracts brokered through Belgian and German traders and then sold to somewhere else? Or are thousands of tons of beans actually shipped to these two countries and then reshipped from there to somewhere else again, without being roasted or processed? You should try to provide some context here. 

I also have trouble with figure 12 and the relevance of "closeness centrality". Are we talking spatial closeness here, as in neighboring countries to exporting countries? Why would such closeness be important or relevant? For example, Burundi and Cameroon have large circles, and what is the significance of that? 

In addition, figures 14 and 16 are frustrating and borderline useless. I like visual presentations, but have a hard time with these. In figure 14, is the cluster of ten nodes on top a representation of European (importing) countries? And fast forward 22 years, why has the shape changed so dramatically in figure 16? I have tried to understand your explanations in the text, but fall short, unfortunately. 

You make some claims in the Conclusions that are hard to defend given what is basically a static analysis of coffee bean trade data from two years, some twenty years apart. For example, you write that "With this analysis, new ways to diagnose a country's problems and thus to develop a political and economic strategy emerge". I find that to be an oversell. What your data seems to show, with or without fancy network analysis, is that the coffee bean trade patterns have changed a bit over the 22-year period. Some producers have become more dominant, and for some unexplained reason two European countries have wedged themselves in as possible intermediaries or transit shipping points. 

It is a different analysis to show why the coffee prices have declined over the past decades, and you are not really getting into an analysis or discussion of macroeconomic factors and developments. For example, it is well known that when Vietnam entered the coffee market in full force about twenty years ago, they caused a severe oversupply that the markets are still reeling from. 

My suggestion is to more closely explain the nature of your data, better explain what the various network analysis tools can show, and present the graphs in a much more user-friendly way, for example, by imposing them on world maps so that readers can see the actual cargo (or value) flows. I would also dedicate more space to a discussion of what has happened with the global coffee market over the past thirty years, instead of claiming too much insight from the network analysis that you have produced. 

It is a fascinating topic that many people would be interested in if they could better understand your arguments for using network analysis here. 

Author Response

We would like to thank Reviewer 1 for all his/her questions. Many of them have made us reflect on our work and by giving you the answers we hope we have clarified some issues and improved the manuscript.

1. In my view, what you have done is taken a dataset of trade figures (imports and exports of green coffee beans, presumably in dollars and not volume) and run these through a network analysis application. Some of the resulting graphs are instructive, while others are quite confusing and don't seem to add much to understanding. It is a bit unclear what your database consists of (which fields it contains), but if it is just exports and imports of a certain HS code, running this through an advanced network program seems like an overkill. However, if your data contains traceable data on intermediate transactions (e.g. beans sold and transiting through intermediate countries), a network analysis would be justified. The nature of your data is not clear, so this should be explained better.

We thank you for this comment. In the last paragraph of the Data section, it was explained that the commodity is coffee, HS6, in US Dollars. But we agree that more details on the data fields were needed. We have now clearly indicated in the text the datasets used for our analysis.

On the other hand, we see that you are not very convinced about the use of complex network tools for the analysis of the international coffee market. We have tried to convince you that complex networks analysis can be a good tool for understanding the market, answering in detail your questions to mitigate those doubts, we hope we have been able to achieve it.

In our opinion, the use of complex network analysis for global visualization is a useful tool, since it allows a single image to see what the market for this commodity is like. Many authors support this idea and apply complex networks analysis to study International Trade networks of different commodities (Basevi and Biggiero 2016, Serrano et al. 2007).

2. What is frustrating for a geographically inclined reader such as me is how you have presented figures 4-7, 12 and 13. I feel all these circles should be implanted on a world map instead of appearing randomly in big spheres. Would not a combined version of figures 10 and 11 tell us all we need to know? Here we can actually see trade relationships and magnitudes.

Sorry for the frustration with our figures, but these representations are the classic figures in graph theory or complex networks. In the latter cases, they are not a random distribution of spheres, the most important nodes are closer to the center of the network, and this representation better shows the topological properties of the network, highlighting the important nodes. In fact, we are more interested in the topological properties of the network than the geolocation of its nodes. On the other hand, Figures 10 and 11 are not the best option to show topological properties because, for example, the Central American countries are too geographically close to each other to have a good view of the interactions, but in these cases, we are more interested in showing that the countries with the greatest intermediation have moved from Central America to Europe, that is why we represent our figures on the world map.

3. I'm not sure how to interpret "betweenness centrality" as long as you have not indicated whether some of the coffee transits between several countries (nodes) before it arrives at the destination. It is mystifying why in 1995 the Central American countries have high betweenness scores, as you would think they would be direct exporters?

The study does not analyze the mobility of coffee internationally but, as indicated in the materials and methods, the data on the commercialization of green coffee which, being a commodity, can be commercialized without physical movement of the product, and even in the future market before the harvest itself.

The text has been included in the article section to clarify the interpretation of this analysis.

Regarding betweenness centrality, a high value of betweenness centrality indicates that countries are well connected inside and outside of other countries in the trade. In 1975, the commercial exchange between the producing countries caused the betweenness centrality to be high among them, for the simple fact of being sellers and buyers for the adjustment of the market. But when the international market has developed, true intermediaries have appeared, that is, countries that serve as buyers and sellers, without being true coffee producers. We have modified the section on betweenness centrality in the manuscript for better understanding.

4. I can sort of understand the high Belgium/Germany betweenness scores in 2017, even though you don't explain why these two countries have become such important intermediaries. In the same vein, does your data reflect monetary flows or the actual physical flows of the coffee beans? In other words, are coffee contracts brokered through Belgian and German traders and then sold to somewhere else? Or are thousands of tons of beans actually shipped to these two countries and then reshipped from there to somewhere else again, without being roasted or processed? You should try to provide some context here.

Coffee trading is a forward trade and purchases are made on the London and New York stock exchanges, as indicated in the article. Therefore, given the nature of this type of economic transactions, which also occurs in other agricultural commodities, the physical presence of coffee beans in these countries is not required.

5. I also have trouble with figure 12 and the relevance of "closeness centrality". Are we talking spatial closeness here, as in neighboring countries to exporting countries? Why would such closeness be important or relevant? For example, Burundi and Cameroon have large circles, and what is the significance of that?

  The text has been included in the article section to clarify the importance of this analysis.

Betweenness and Closeness centralities are both centrality measures using the shortest path between nodes. Betweenness centrality is generally regarded as a measure of others’ dependence on a node, and therefore as a measure of potential control. Closeness centrality is usually interpreted either as a measure of access efficiency or of independence from potential control by intermediaries. But both measures are topological measures, not geographical measures.

These figures once again remark the idea that in 1995, the market was dominated by producers, but as the market volume developed, intermediaries, have appeared, causing the nodes of high closeness centrality to go from Cote d'Ivoire and Cameroon to Germany, Belgium, or Luxembourg.

6. In addition, figures 14 and 16 are frustrating and borderline useless. I like visual presentations, but have a hard time with these. In figure 14, is the cluster of ten nodes on top a representation of European (importing) countries? And fast forward 22 years, why has the shape changed so dramatically in figure 16? I have tried to understand your explanations in the text, but fall short, unfortunately.

We agree with the reviewer that these figures may not be very useful, but they are added in the text first for completeness and second to show the complete view of modularity in order to later explain Figures 15 and 17, which we think are interesting to understand the members of the different classes.

Figure 14 and Figure 15 are related and keep the same color code. The cluster of the ten nodes corresponds to modularity class 3 in Figure 15. Text has been included in the figure to clarify it.

This cluster is made up of some African countries along with the United Kingdom and Switzerland and a mix of Costa Rica, China, and Hong Kong. The fact that it is separated from the rest of the clusters probably indicates that this cluster exhibits differential features compared to the others, although a more in-depth analysis would be needed to explain those differences. The modification of the coffee market shown may be due to different factors: macroeconomic, geopolitical, etc., which require different and extensive specific studies on each aspect. But these ideas are beyond the scope of this article. The objective is to show how the connections and protagonists of the countries that make up the green coffee commercial network are changing.

7. You make some claims in the Conclusions that are hard to defend given what is basically a static analysis of coffee bean trade data from two years, some twenty years apart. For example, you write that "With this analysis, new ways to diagnose a country's problems and thus to develop a political and economic strategy emerge". I find that to be an oversell. What your data seems to show, with or without fancy network analysis, is that the coffee bean trade patterns have changed a bit over the 22-year period. Some producers have become more dominant, and for some unexplained reason two European countries have wedged themselves in as possible intermediaries or transit shipping points.

It is possible that the conclusions tend to exaggerate the importance of the results, but we would like to clarify that we have not only studied the two years that we show at work. We have done analysis in many of the years in the period between 1975 and 2004, but in order not to overwhelm the reader with too much data we show the beginning and the end of evolution.

What we do not agree too much with the reviewer is that the market change is small, we think that the change can have significant implications at the economic level, especially in small producing countries, generating sustainability problems. Nor can we share the idea that the analysis of complex networks does not provide much utility to the analysis, although we do agree that the analysis is fancy. We believe that the fact that with a single image of the market we can see all the interactions and that with two we can see how the market has evolved can be very useful to attract attention to the evolution of the market. In addition, representing these images, from different points of view of complex networks, for example looking at their in-degree or out- degree or betweenness centrality, we can highlight importing, exporting countries, or intermediaries.

We have modified the text of the conclusions to reinforce these ideas.

8. It is a different analysis to show why the coffee prices have declined over the past decades, and you are not really getting into an analysis or discussion of macroeconomic factors and developments. For example, it is well known that when Vietnam entered the coffee market in full force about twenty years ago, they caused a severe oversupply that the markets are still reeling from.

We fully agree that more exhaustive and detailed analyses of the evolution of the coffee market are necessary, such as those made by Ponte (2002), who makes a complete study of the coffee price problem. In our article, we only intended to give a global vision of the evolution of the market using different tools and show the possibilities that these tools offer to carry out more in- depth studies.

The objective of this article is to analyze the evolution, over time, of the importance of countries in the green coffee market, in order to visualize what has happened, especially with the producing countries in the last 20 years.

9. My suggestion is to more closely explain the nature of your data, better explain what the various network analysis tools can show, and present the graphs in a much more user- friendly way, for example, by imposing them on world maps so that readers can see the actual cargo (or value) flows. I would also dedicate more space to a discussion of what has happened with the global coffee market over the past thirty years, instead of claiming too much insight from the network analysis that you have produced.

In our article, we only intend to give a global vision of the evolution of the market using different tools and show the possibilities that these tools offer to carry out more in-depth studies. We think that the ideas that the reviewer proposes may be very interesting but they exceed the scope of this article and they can be helpful for a further article.

10. It is a fascinating topic that many people would be interested in if they could better understand your arguments for using network analysis here.

Thank you for thinking that the analysis of complex networks can be a fascinating topic, we hope that with the modifications and clarifications that we have made thanks to the comments of the reviewers and the editor, it would be possible to appreciate some of those fascinating advantages. We understand that there is more work to be done in this type of analysis but we think that they could be topics for future studies that could be done by our group or that the article could help inspire other researchers.

Reviewer 2 Report

This paper aimed to examine the dynamics and evolution of the international green coffee market and the re-distribution of value in the coffee supply chain. The topic of this article is of interest to wide readers. The paper's argument is built on an appropriate base of theory and the methods employed in the paper are appropriate, however, the paper does not have any sustainability dimension, the word “Sustainability” is not mentioned anywhere in the article.

Author Response

1. This paper aimed to examine the dynamics and evolution of the international green coffee market and the re-distribution of value in the coffee supply chain. The topic of this article is of interest to wide readers. The paper's argument is built on an appropriate base of theory and the methods employed in the paper are appropriate, however, the paper does not have any sustainability dimension, the word “Sustainability” is not mentioned anywhere in the article.

We completely agree with the reviewer. We have included some paragraphs in the manuscript introducing the sustainability dimension.

Round 2

Reviewer 1 Report

Thank you for adding several explanatory comments to make the article more accessible and ease the understanding for readers who might not use network analysis and jargon on a daily basis. I believe you could even include more of the responses you sent me as explanatory (context-giving) content in your final paper, as it helped me quite a bit. It would not hurt to make it as reader-friendly as possible, as it also makes your own argumentation and logic appear clearer.  

I believe this is an interesting and worthy contribution to research and scholarly work on the international coffee market. Happy New Year!

Author Response

Thank you very much for the comments to the article. We believe that thanks to all of them the article has improved substantially. We have added some of the comments in the new manuscript as requested. Happy New Year too.

Reviewer 2 Report

I am satisfied with the revised version of the manuscript as the authors adequately responded to all reviewers’ comments. The manuscript has considerably improved and so far, it is endorsed for final publication.

Author Response

Thank you very much for the comments to the article. We believe that thanks to all of them the article has improved substantially.